# LearniBridge: Learnable Calibration of Feature Caching for Diffusion Models Acceleration

Xuyue Huang [1]    Zhe Chen [2]    Wang Shen [2]    Xiao-Ping Zhang [1]

## Abstract

Diffusion Transformers (DiTs) have driven substantial progress in image and video generation but suffer from prohibitive computational costs. Feature caching accelerates inference by reusing intermediate representations. Existing methods rely on historical features for implementation simplicity, yet suffer from severe error accumulation at high acceleration ratios. To address this limitation, we investigate the nature of the requisite feature correction. We demonstrate that the optimal calibration update is characterized by a shared low-rank subspace across diverse prompts. Guided by this structural insight, we propose *LearniBridge*, a learnable calibration mechanism for feature caching that bridges multiple timesteps through lightweight LoRA updates. This mechanism enables effective calibration requiring only 3–5 training samples. Extensive experiments on image and video generation show that *LearniBridge* achieves up to $5.87\times$, $5.75\times$, and $4.10\times$ acceleration on FLUX, HunyuanVideo, and WAN 2.1, respectively. On WAN 2.1, it improves VBench by 1.28% over the previous SOTA at $4.10\times$ acceleration. Our code is available at https://github.com/Iiiiiiirene/LearniBridge.

## 1. Introduction

Diffusion models (Ho et al., 2020; Dhariwal & Nichol, 2021) have rapidly advanced generative modeling, achieving state-of-the-art performance in image and video generation (Rombach et al., 2022; Blattmann et al., 2023). Recent work introduces Diffusion Transformers (DiTs) (Peebles & Xie,

[1]Shenzhen Ubiquitous Data Enabling Key Laboratory, Shenzhen International Graduate School, Tsinghua University, Shenzhen, China [2]Central Media Technology Institute, Huawei Technologies Co., Ltd., Shenzhen, China. Correspondence to: Xiao-Ping Zhang <xpzhang@ieee.org>.

*Proceedings of the 43rd International Conference on Machine Learning*, Seoul, South Korea. PMLR 306, 2026. Copyright 2026 by the author(s).

2023; Esser et al., 2024; Chen et al., 2024b) to further enhance generation quality. However, these improvements come at the expense of high computational demands, which limit the practical applicability.

Various acceleration strategies (Lu et al., 2022; Li et al., 2025) have been proposed to address these challenges. Among them, feature caching (Liu et al., 2025a; Zhao et al., 2025; Yuan et al., 2024; Zou et al., 2025; Zhou et al., 2025) effectively reduces inference costs by reusing intermediate representations from earlier timesteps. Methods such as DeepCache(Ma et al., 2024) and FORA (Selvaraju et al., 2024) exploit the similarity between adjacent timesteps for direct reuse. To handle longer caching intervals, TaylorSeer (Liu et al., 2025b) employs Taylor-series expansions over multi-step histories to predict future feature features. However, TaylorSeer requires more cached features and relies on the assumption of smooth, higher-order differentiable feature trajectories that may not always hold.

Existing methods rely on historical features for implementation simplicity, yet incur severe error accumulation at high acceleration ratios. This raises a critical question: Is there an underlying structure within these caching errors that enables efficient, learnable correction? In this paper, by analyzing the spectral properties of feature correction matrices, we observe that the requisite updates consistently reside in a low-dimensional subspace. This theoretically justifies that the calibration weights are naturally constrained by a low-rank structure. Furthermore, we find that these low-rank subspaces remain remarkably consistent across diverse prompts, demonstrating a robust prompt-invariant structure.

Motivated by these observations, we introduce *LearniBridge*, a learnable calibration mechanism for cached features, implemented via lightweight LoRA updates that bridge multiple timesteps. The method consists of two phases: training and inference. In the training phase, we optimize LoRA-based calibration weights to align cached features with their full-computation representations. In the inference phase, full computation at the target timestep is bypassed, requiring only the inference of the LoRA-augmented final Transformer block. Due to the inherent low-rank nature of these calibration weights, our approach is highly parameter-

efficient. Furthermore, the prompt-invariant structure enables *LearniBridge* to generalize broadly after training on minimal samples. This design enables *LearniBridge* to bridge multiple timesteps with minimal computational overhead.

Our main contributions are summarized as follows:

- **Prompt-Invariant Low-Rank Structure:** We demonstrate that the optimal calibration updates in DiT linear layers reside in a prompt-invariant, low-dimensional subspace. This finding provides a principled foundation for effective calibration.

- **LearniBridge:** We propose a LoRA-based framework that calibrates cached features to reconstruct future-step representations. By using only 3-5 training prompts and lightweight low-rank updates, it enables efficient integration with negligible computational overhead.

- **Outstanding performance:** *LearniBridge* achieves up to $5.87\times$, $5.75\times$, and $4.10\times$ acceleration on FLUX, HunyuanVideo, and Wan 2.1, respectively, while maintaining high generation quality across image and video generation tasks.

## 2. Related Work

Diffusion models (Ho et al., 2020; Sohl-Dickstein et al., 2015) have achieved remarkable success in generative tasks. Early works predominantly employed U-Net structures (Ronneberger et al., 2015), but struggled with scalability in larger models. This changed with the Diffusion Transformers (DiTs) (Peebles & Xie, 2023), which resolved these constraints and enabled state-of-the-art performance in multiple domains (Chen et al., 2024a;b; Yang et al., 2025; Zheng et al., 2024). However, sequential sampling leads to persistently high inference costs. To address this, acceleration strategies have been extensively studied and are generally divided into Sampling Timestep Reduction and Denoising Network Acceleration.

### 2.1. Diffusion Model Acceleration

Extensive research focuses on accelerating diffusion models by either minimizing sampling steps or enhancing per-step efficiency. DDIM (Song et al., 2021) introduced a deterministic sampling paradigm that preserves generation fidelity with fewer iterations, a framework subsequently refined by the DPM-Solver series (Lu et al., 2022) through higher-order ODE solvers. Consistency Models (Song et al., 2023) further establish self-consistent noise-to-data mappings, enabling one- or few-step generation. Additionally, model distillation (Salimans & Ho; Meng et al., 2023) compresses multi-step samplers into efficient student models that require significantly fewer denoising iterations. To further alleviate the per-step computational burden, model quantization (Kim et al., 2025; Li et al., 2023; Shang et al., 2023) and structural pruning (Fang et al., 2023; Zhu et al., 2024) compress the diffusion backbones. Beyond these approaches, feature caching represents a highly efficient acceleration paradigm due to its minimal training overhead and model-agnostic nature.

### 2.2. Feature Caching

Caching-based methods exploit the strong temporal coherence of intermediate activations to skip redundant computation across diffusion timesteps. DeepCache (Song et al., 2023) initially developed for U-Net architectures, reuses features across multiple steps. Extending this concept to Transformer-based architectures, FORA (Selvaraju et al., 2024) implements fundamental module-level output caching specifically for Diffusion Transformers (DiTs). DiTFastAttn (Yuan et al., 2024) further reduces costs by sharing attention outputs across spatial dimensions, time, and conditional branches. To maintain synthesis quality, ToCa (Zou et al., 2025) incorporates dynamic feature updates to mitigate information loss caused by feature aging. In terms of cache decision mechanisms, TeaCache (Liu et al., 2025a) introduces a calibrated polynomial estimator to predict output changes from input differences. TaylorSeer (Liu et al., 2025b) advances the paradigm from simple "replication" to "prediction" via a Taylor-series-inspired extrapolation scheme, significantly enhancing generation quality during long-range skipping. Despite these advancements, existing methods predominantly rely on historical features for implementation simplicity, which inevitably incurs severe error accumulation at high acceleration ratios.

## 3. Method

### 3.1. Preliminary

**Diffusion Models.** Diffusion models formulate generative modeling as learning to invert a gradual noising process. Starting from clean data $x_0 \sim q(x)$, a forward diffusion process constructs a sequence $\{x_t\}_{t=1}^T$ by progressively adding Gaussian noise:

$$x_t = \sqrt{\alpha_t}\, x_{t-1} + \sqrt{1 - \alpha_t}\, z_t, \quad z_t \sim \mathcal{N}(0, I), \quad (1)$$

where $\alpha_t \in (0, 1]$ controls the signal-to-noise ratio at each timestep. For appropriately chosen $\{\alpha_t\}_{t=1}^T$, the marginal distribution of $x_T$ approaches an isotropic Gaussian.

The generative model parameterizes the reverse process using a neural network that defines the conditional distributions:

$$p_\theta(x_{t-1} \mid x_t) = \mathcal{N}(x_{t-1}; \mu_\theta(x_t, t), \Sigma_\theta(x_t, t)), \quad (2)$$

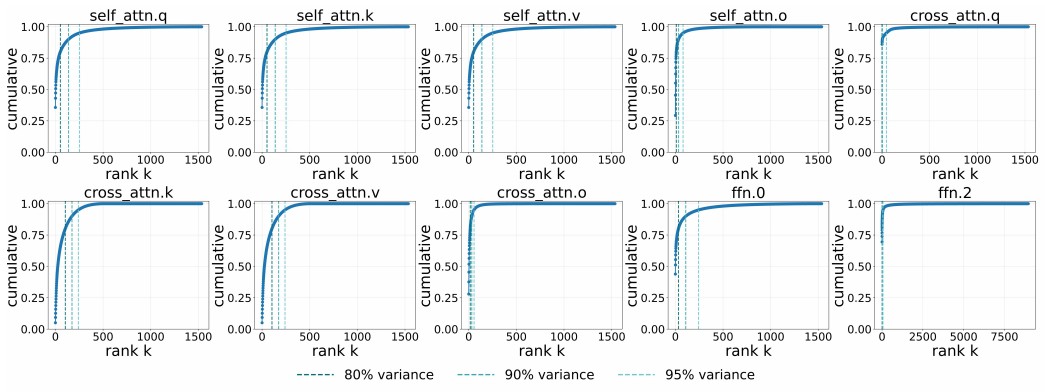

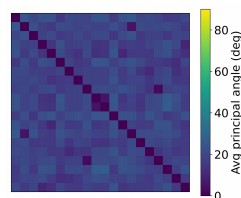

*Figure 2.* Small angles between updates from disjoint prompt groups verify that the correction pattern is prompt-invariant.

*Figure 1.* SVD analysis of the aggregated input matrix $X_t^l$ across 100 distinct prompts. The singular values exhibit a rapid decay, indicating that spectral energy is concentrated in a few principal components. This implies that $X_t^l$ possesses an intrinsic low-rank structure, constraining the optimal correction $\Delta W^l$ to be low-rank.

and samples are obtained by iteratively applying these reverse transitions from $t = T$ down to $t = 1$. Since this procedure requires evaluating the backbone network at every timestep, diffusion models typically incur substantial computational cost during generation.

**Diffusion Transformer Architecture.** The Diffusion Transformers (DiTs) adopts a hierarchical architecture $G = g_1 \circ \cdots \circ g_L$. Each block is defined as $g_l = F_{\text{SA}}^l \circ F_{\text{CA}}^l \circ F_{\text{MLP}}^l$, indicating that the input is sequentially processed by a feedforward (MLP) module, a cross-attention module, and a self-attention module. Each module is implemented in residual form as $F_\alpha^l(x) = x + \text{AdaLN}_\alpha^l\big(f_\alpha^l(x)\big)$ for $\alpha \in \{\text{SA}, \text{CA}, \text{MLP}\}$.

The self-attention module computes the query, key, and value projections $Q = xW_Q^l$, $K = xW_K^l$, and $V = xW_V^l$, followed by scaled dot-product attention $\text{Attn}(x) = \text{softmax}\big(QK^\top/\sqrt{d_h}\big)V$ and an output projection $f_{\text{SA}}^l(x) = \text{Attn}(x)W_O^l$. The cross-attention module follows an identical formulation with its own projection matrices. The MLP module consists of two linear layers with a nonlinear activation, expressed as $f_{\text{MLP}}^l(x) = \sigma\big(xW_1^l\big)W_2^l$. Together, these components define the standard DiT backbone that is applied consistently across diffusion timesteps.

**Feature Caching.** Feature caching reduces computational cost by approximating block outputs across diffusion timesteps. For block $l$ at timestep $t$, let $F^l\big(x_t^l\big)$ denote its output, and let $c_s^l = F^l\big(x_s^l\big)$ represent the cached output at a reference timestep $s$. Given a set of reference timesteps $S_{t,k} \subseteq \{t, \ldots, t - m\}$ with $k \in \{1, \ldots, m\}$, a general caching scheme approximates the block output at timestep $t - k$ as:

$$\hat{F}^l\big(x_{t-k}^l\big) = \Phi^l\Big(\{c_s^l\}_{s \in S_{t,k}}, k\Big), \qquad (3)$$

where $\Phi^l$ specifies the rule by which cached features are combined to construct the approximation. Different caching methods instantiate $\Phi^l$ using different strategies, such as direct feature reuse or higher-order extrapolation.

**Low-Rank Adaptation.** Low-Rank Adaptation (LoRA) introduces trainable low-rank matrices into linear layers to enable parameter-efficient fine-tuning (Hu et al., 2022). For a linear transformation with weight $W^l \in \mathbb{R}^{d_{\text{in}} \times d_{\text{out}}}$, LoRA augments the weight with a low-rank update:

$$\Delta W^l = B^l A^l, \qquad (4)$$

where $A^l \in \mathbb{R}^{r \times d_{\text{out}}}$ and $B^l \in \mathbb{R}^{d_{\text{in}} \times r}$, with $r \ll \min(d_{\text{in}}, d_{\text{out}})$. During adaptation, the effective weight becomes $W^l + \Delta W^l$, while the base weight $W^l$ remains frozen. Only the low-rank factors $A^l$ and $B^l$ are updated, which substantially reduces the number of trainable parameters and allows the adapted weights to be stored and applied in a modular manner.

### 3.2. Prompt-Invariant Low-Rank Calibration Update

Let $f^l(x_{t-k}^l)$ denote the output of a specific linear layer at the reference timestep $t - k$, where $x_{t-k}^l$ is the corresponding input. We investigate the feature shift between the layer outputs given the previous input $x_t^l$ versus the current input $x_{t-k}^l$. For each training sample $i$, we define the cross-timestep residual as:

$$e_{t \to t-k,i}^l \triangleq f^l\big(x_{t-k,i}^l\big) - f^l\big(x_{t,i}^l\big). \qquad (5)$$

We aggregate these residuals and the corresponding inputs from all $N$ samples into matrices $E_{t \to t-k}^l$ and $X_t^l$, respectively:

$$E_{t \to t-k}^l = \big[e_{t \to t-k,1}^l, \ldots, e_{t \to t-k,i}^l, \ldots, e_{t \to t-k,N}^l\big], \qquad (6)$$

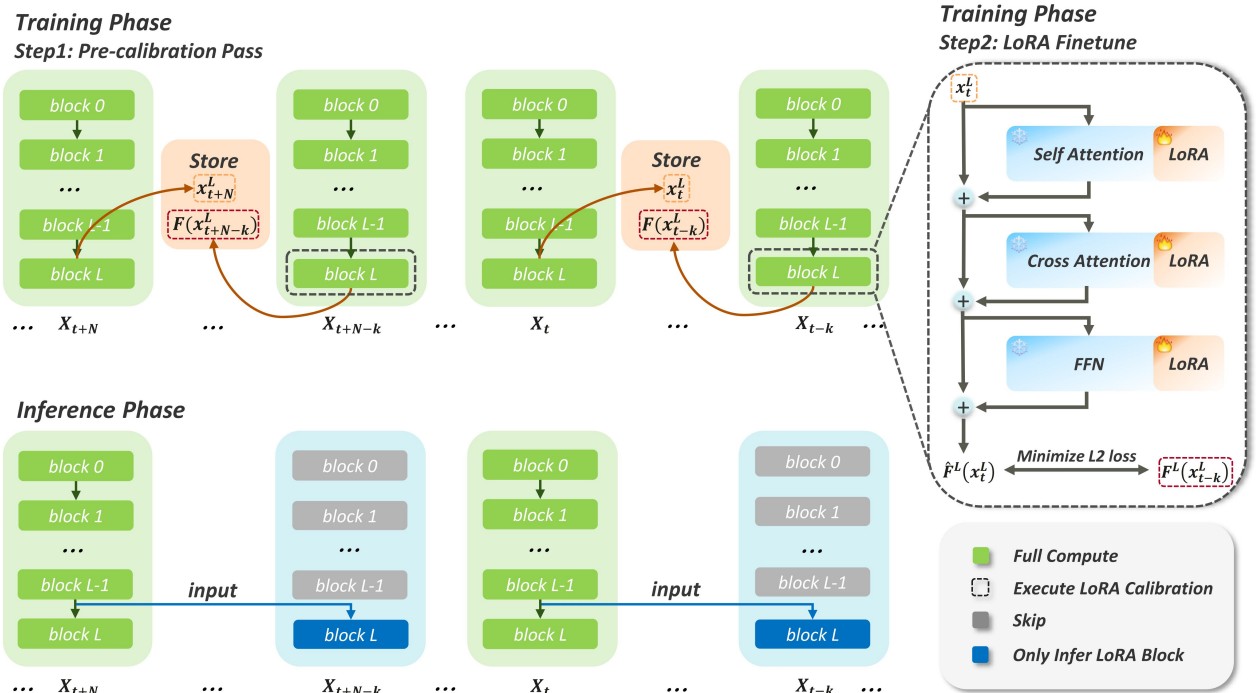

*Figure 3.* Overview of *LearniBridge*. Our method consists of a training phase and an inference phase. During the training phase, a pre-calibration pass performs full computation at all timesteps, recording the final-block input $x_t^L$ and the corresponding ground-truth outputs $F^L(x_{t-k}^L)$ for calibrated timesteps. In the LoRA Finetune process, LoRA adapters are trained in the final block to map the cached input $x_t^L$ to the corresponding full-computation output $F^L(x_{t-k}^L)$. During the inference phase, full computation at the target timesteps is skipped, only infer the LoRA-augmented final Transformer block.

$$X_t^l = \left[ x_{t,1}^l, \ldots, x_{t,i}^l, \ldots, x_{t,N}^l \right]. \tag{7}$$

Let $\Delta W^l$ represent the adaptive linear correction introduced to compensate for this discrepancy. We model the relationship as $E_{t \to t-k}^l \approx \Delta W^l X_t^l$. Minimizing the reconstruction error yields the closed-form solution:

$$\Delta W^l = E_{t \to t-k}^l (X_t^l)^\dagger, \tag{8}$$

where $(X_t^l)^\dagger$ denotes the Moore-Penrose pseudo-inverse.

We next analyze the spectral properties of the input matrix $X_t^l$ via Singular Value Decomposition (SVD):

$$X_t^l = U\Sigma V^\top, \tag{9}$$

where $U$ and $V$ are orthogonal matrices, and $\Sigma$ is a diagonal matrix of singular values. Substituting the pseudo-inverse expression $(X_t^l)^\dagger = V\Sigma^+ U^\top$ into the solution for $\Delta W^l$, we obtain:

$$\Delta W^l = E_{t \to t-k}^l V\Sigma^+ U^\top. \tag{10}$$

As illustrated in Figure 1, we constructed $X_t^l$ using samples from 100 distinct prompts. The singular values of $X_t^l$ exhibit a rapid decay, indicating that the spectral energy is concentrated in a few principal components. This empirical observation suggests that $X_t^l$ possesses an intrinsic low-rank structure. Given that the rank of a matrix product

is constrained by the minimum rank of its factors, it follows that:

$$\text{rank}(\Delta W^l) \leq \text{rank}((X_t^l)^\dagger) = \text{rank}(X_t^l) \leq r. \tag{11}$$

This implies that the optimal correction $\Delta W^l$ is inherently low-rank.

To verify the universality of the update direction, we randomly partition 100 prompts into 20 disjoint groups and compute the optimal $\Delta W^l$ for each group. For each $\Delta W^l$, we perform SVD to obtain its principal subspace, and Figure 2 visualizes the pairwise angles between these subspaces across different prompt groups. The consistently small angles reveal a strong structural similarity between these updates, implying that the required feature correction follows a universal pattern, independent of the specific input prompts.

### 3.3. LoRA-Based Calibration Architectures

Let $F^l(x_{t-k}^l)$ denote the output of a specific block at the reference timestep $t - k$. Leveraging the low-rank update structure, the output at a later timestep can be approximated from the input at an earlier timestep as:

$$F^l(x_{t-k}^l; W^l) \approx F^l(x_t^l; W^l + \Delta W^l). \tag{12}$$

This relation indicates that a suitably parameterized low-

*Table 1.* Quantitative comparison in text-to-image generation on DrawBench with FLUX.1-dev.

| Method FLUX.1-dev | Acceleration | | | | Quality Metrics | | | | |
|---|---|---|---|---|---|---|---|---|---|
| | Latency(s)↓ | Speed↑ | FLOPs(T)↓ | Speed↑ | ImageReward↑ | CLIP↑ | PSNR↑ | SSIM↑ | LPIPS↓ |
| Original | 27.32 | 1.00× | 3719.5 | 1.00× | 0.9885 | 0.8102 | – | – | – |
| 22% steps | 6.00 | 4.55× | 817.5 | 4.55× | 0.8669 | 0.8130 | 25.9587 | 0.6720 | 0.3691 |
| ToCA ($N$=9) (Zou et al., 2025) | 6.88 | 3.97× | 854.4 | 4.35× | 0.8352 | 0.8045 | 27.9813 | 0.7012 | 0.3155 |
| TeaCache ($\delta$=0.8) (Liu et al., 2025a) | 6.60 | 4.14× | 892.0 | 4.17× | 0.8975 | 0.8103 | 28.6508 | 0.7350 | **0.2538** |
| TaylorSeer ($N$=5,$O$=2) (Liu et al., 2025b) | 6.20 | 4.41× | 817.5 | 4.55× | 0.9359 | **0.8164** | 29.8558 | 0.7625 | 0.2697 |
| **LearniBridge** ($N$=5) | 6.27 | 4.36× | 839.6 | 4.43× | **0.9590** | 0.8128 | **30.1525** | **0.7879** | 0.2682 |
| ToCA ($N$=10) (Zou et al., 2025) | 5.78 | 4.73× | 714.7 | 5.20× | 0.7998 | 0.7956 | 26.9854 | 0.6390 | 0.3702 |
| TeaCache ($\delta$=1) (Liu et al., 2025a) | 5.66 | 4.83× | 743.6 | 4.89× | 0.8398 | 0.8060 | 27.0821 | 0.6996 | 0.3702 |
| TaylorSeer ($N$=6,$O$=2) (Liu et al., 2025b) | 5.47 | 4.99× | 745.4 | 4.99× | 0.9033 | 0.8094 | 28.7006 | 0.7191 | 0.3109 |
| **LearniBridge** ($N$=6) | 5.61 | 4.87× | 759.1 | 4.90× | **0.9133** | **0.8364** | **29.7491** | **0.7407** | **0.3021** |
| ToCA ($N$=12) (Zou et al., 2025) | 4.65 | 5.87× | 628.3 | 5.92× | 0.7019 | 0.7826 | 26.4802 | 0.5856 | 0.3928 |
| TeaCache ($\delta$=1.4) (Liu et al., 2025a) | 4.56 | 5.99× | 603.8 | 6.16× | 0.7252 | 0.8026 | 26.5802 | 0.6338 | 0.3928 |
| TaylorSeer ($N$=8,$O$=2) (Liu et al., 2025b) | 4.48 | 6.10× | 596.1 | 6.24× | 0.8212 | 0.8041 | 26.8228 | 0.6750 | 0.3647 |
| **LearniBridge** ($N$=8) | 4.43 | 6.17× | 599.9 | 6.20× | **0.8308** | **0.8164** | **28.3464** | **0.6870** | **0.3549** |

rank adapter can reproduce the skipped-timestep representation using only cached features.

As illustrated in Figure 3, *LearniBridge* implements this insight by applying a lightweight residual correction to the final Transformer block $g_L$ (composed as $g_l = F_{SA}^l \circ F_{CA}^l \circ F_{MLP}^l$). This mechanism compensates for the temporal feature shift by modeling it as a learnable update. Restricting LoRA to $g_L$ minimizes trainable parameters and memory footprint while preserving the backbone architecture. Since no auxiliary blocks are executed, this design incurs negligible inference latency, yielding a plug-and-play calibration module.

For any linear transformation $W^l$, LoRA introduces a low-rank update:

$$\Delta W^l = B^l A^l, \qquad r \ll \min(d_{\text{in}}, d_{\text{out}}), \qquad (13)$$

and replaces $W^l$ with $W^l + \Delta W^l$ while keeping the base weight $W^l$ itself frozen. *LearniBridge* applies this parameterization exclusively to the final block $g_L$, introducing low-rank updates $\Delta W_Q^L, \Delta W_K^L, \Delta W_V^L, \Delta W_O^L, \Delta W_1^L$, and $\Delta W_2^L$ to all linear layers within the block.

**Training Phase**
During pre-calibration pass, we collect a small set of prompts (3–5) and run full diffusion trajectories. For each fully computed timestep $t$, we record the input to the final block, denoted as $x_t^L$. For timesteps $x_{t-k}^L$ that will be calibrated during inference, we record corresponding ground-truth outputs $F^L(x_{t-k}^L)$.

During LoRA finetune process, the LoRA parameters in $g_L$ are trained while keeping all base weights frozen. The objective is to ensure that the LoRA-augmented final block, when taking the cached feature $x_t^L$ as input, can well approximate the target output $F^L(x_{t-k}^L)$ obtained under full computation. Over all training pairs $(t, t-k)$, we minimize:

$$\mathcal{L}_{\text{LearniBridge}} = \sum_i \left\| \hat{F}^L\left(x_{t,i}^L\right) - F^L\left(x_{t-k,i}^L\right) \right\|_2^2. \quad (14)$$

Here, $\hat{F}^L(x_{t,i}^L)$ serves as the output of LoRA-augmented final block. By modeling the temporal shift, we reconstruct representations at skipped timesteps directly from the cached features.

**Inference Phase**
During inference, the model periodically executes full computation at intervals of $N$ timesteps and caches the corresponding input to the final block, denoted as $x_t^L$. For a target timestep $t-k$, we retrieve the cached feature $x_t^L$ from the nearest full-compute step $t$. Instead of recomputing all blocks $g_1, \ldots, g_{L-1}$ at timestep $t-k$, *LearniBridge* directly feeds this cached feature $x_t^L$ into the LoRA-augmented final block $g_L$, producing an approximation $\hat{F}^L(x_t^L)$ of the full computation output. This substitution allows the model to bypass earlier blocks, relying on the trained low-rank correction to recover cross-timestep feature evolution.

## 4. Experiments

### 4.1. Experiment Settings

**Model Configurations.** Experiments are conducted on three state-of-the-art visual generative models: the text-to-image generation model FLUX.1-dev (Black Forest Labs, 2024), and the text-to-video generation models Hunyuan-Video (Kong et al., 2024) and WAN 2.1-1.3B (Wan et al., 2025). All images and videos used for both quantitative and qualitative evaluation are generated on Ascend 910B devices.

*Table 2.* Quantitative comparison in text-to-video generation for HunyuanVideo on VBench.

| Method HunyuanVideo | Acceleration | | | | Quality Metrics | | | |
|---|---|---|---|---|---|---|---|---|
| | Latency(s)↓ | Speed↑ | FLOPs(T)↓ | Speed↑ | VBench(%) | PSNR↑ | SSIM↑ | LPIPS↓ |
| Original | 617.79 | 1.00× | 29773.0 | 1.00× | 80.93 | − | − | − |
| 22% steps | 135.78 | 4.55× | 6550.1 | 4.55× | 78.88 | 15.09 | 0.5669 | 0.3997 |
| TeaCache ($\delta$=0.3) (Liu et al., 2025a) | 167.88 | 3.68× | 7794.0 | 3.82× | 80.07 | 18.36 | 0.7155 | 0.2629 |
| TaylorSeer ($N$=4, $O$=1) (Liu et al., 2025b) | 161.30 | 3.83× | 7733.2 | 3.85× | 80.74 | 19.47 | 0.6686 | 0.3096 |
| **LearniBridge** ($N$=4) | 164.74 | 3.75× | 7876.5 | 3.78× | **80.84** | **20.55** | **0.7508** | **0.2314** |
| ToCa ($N$=5) (Zou et al., 2025) | 149.94 | 4.12× | 7005.4 | 4.25× | 79.25 | 18.13 | 0.6002 | 0.3885 |
| TeaCache ($\delta$=0.4) (Liu et al., 2025a) | 130.89 | 4.72× | 6151.5 | 4.84× | 79.40 | 16.91 | 0.6649 | 0.3372 |
| TaylorSeer ($N$=5, $O$=1) (Liu et al., 2025b) | 132.86 | 4.65× | 5966.5 | 4.99× | **80.13** | 18.28 | 0.6121 | 0.3722 |
| **LearniBridge** ($N$=5) | 125.31 | 4.93× | 5966.5 | 4.99× | 79.93 | **18.80** | **0.7396** | **0.2349** |
| TaylorSeer ($N$=7, $O$=1) (Liu et al., 2025b) | 103.30 | 5.98× | 4771.3 | 6.24× | 79.14 | 17.77 | 0.6122 | 0.4232 |
| **LearniBridge** ($N$=7) | 100.45 | 6.15× | 4794.4 | 6.21× | **79.51** | **17.84** | **0.6504** | **0.3780** |

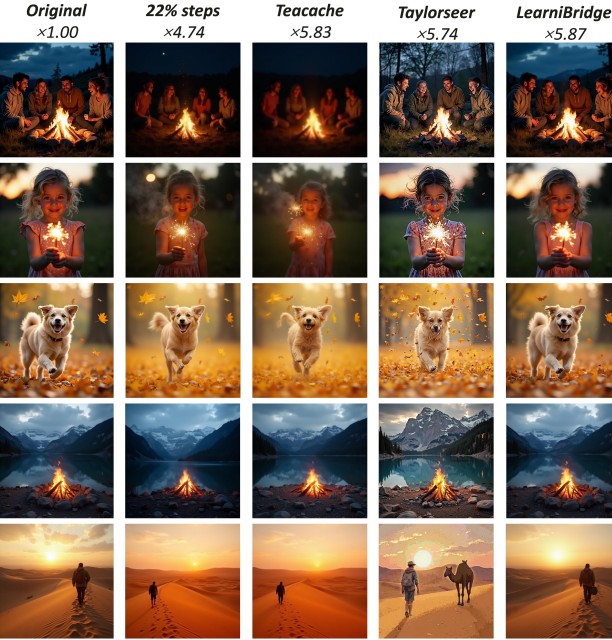

*Figure 4.* Detailed visualization results for different acceleration methods on FLUX.1-dev. Existing methods suffer from severe content deviation, blurring artifacts, or abnormal color contrast at high speedup, whereas *LearniBridge* maintains high content fidelity and superior visual quality even at nearly 6× acceleration.

**Evaluation and Metrics.** For text-to-image generation, we generate 200 images using prompts from the DrawBench benchmark (Saharia et al., 2022). Image quality and text–image alignment are evaluated using ImageReward (Xu et al., 2023) and CLIP Score (Hessel et al., 2021). For text-to-video generation, we produce a total of 4,730 videos by generating five samples for each of the 946 prompts. Model performance is comprehensively evaluated using the VBench (Huang et al., 2024) framework. Additionally, the fidelity of the generated outputs with respect to the original results is quantitatively assessed using PSNR, SSIM (Wang

et al., 2004), and LPIPS (Zhang et al., 2018).

### 4.2. Text-to-Image Generation

**Quantitative Study.** As shown in Table 1, we compare *LearniBridge* with existing acceleration methods, including ToCa, TeaCache, and TaylorSeer, on the FLUX.1-dev model. Under moderate acceleration, *LearniBridge* achieves a 4.36× speedup while maintaining strong semantic and visual quality, with IR (0.9590 ↑), CLIP (0.8128 ↑), PSNR (30.1525 ↑), and SSIM (0.7879 ↑). It outperforms ToCa, TaylorSeer and remains competitive with TeaCache across most metrics. At a 4.98× speedup, *LearniBridge* achieves the highest CLIP score (0.8364 ↑) while maintaining strong perceptual quality. Even under aggressive acceleration at 5.87×, *LearniBridge* preserves highest IR (0.8308 ↑), indicating superior robustness as the acceleration factor increases.

**Qualitative Study.** Figure 4 presents a visual comparison between *LearniBridge* and baseline methods on FLUX.1-dev. When the speedup approaches 6×, TeaCache exhibits significant content deviation from the original outputs, accompanied by noticeable blurring artifacts. TaylorSeer produces results with low similarity to the original images and suffers from abnormal color contrast. In contrast, *LearniBridge* preserves high content consistency with the original outputs while maintaining substantially better visual fidelity, demonstrating its effectiveness under large timestep skipping.

### 4.3. Text-to-Video Generation

**Quantitative Study.** As shown in Table 2, we compare *LearniBridge* with existing acceleration methods, including ToCa, TeaCache, and TaylorSeer, on the HunyuanVideo model. Under moderate acceleration, *LearniBridge* achieves a 3.75× speedup with a VBench score of 80.84, outperforming TeaCache and TaylorSeer at comparable speedup

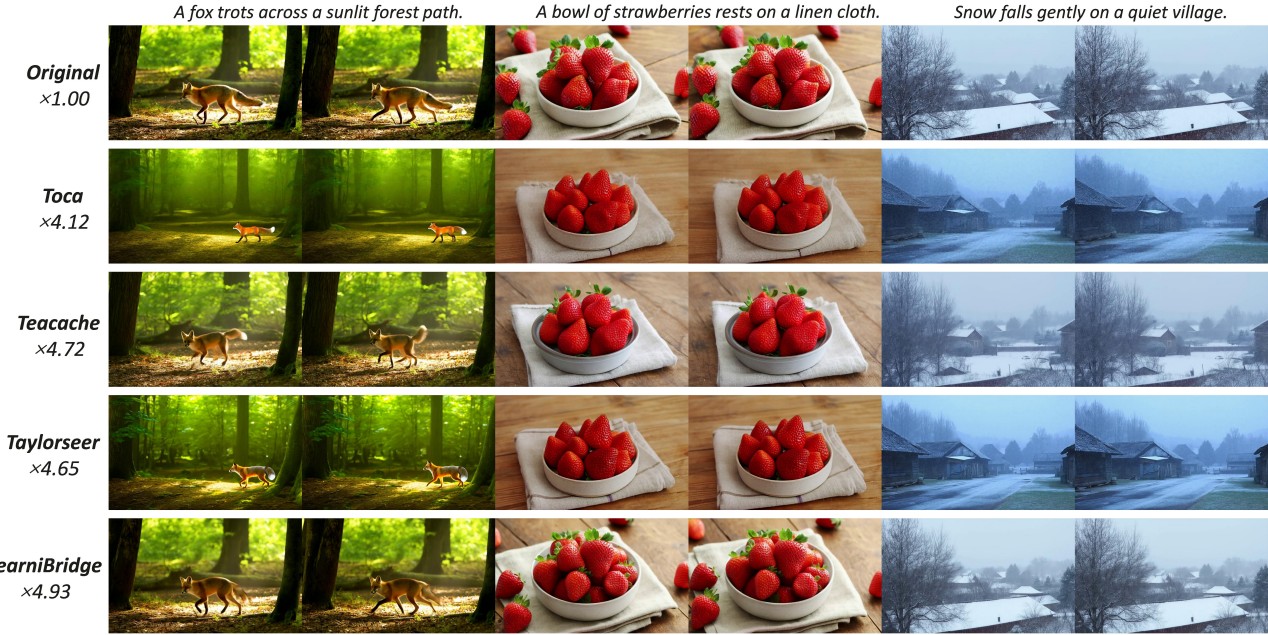

*A fox trots across a sunlit forest path.*  *A bowl of strawberries rests on a linen cloth.*  *Snow falls gently on a quiet village.*

*Figure 5.* Visualization of different acceleration methods on HunyuanVideo. While achieving higher acceleration ratios, other methods exhibit issues such as motion detail loss, content deviation, visual quality degrade. In contrast, our method maintains high-quality generation without these problems.

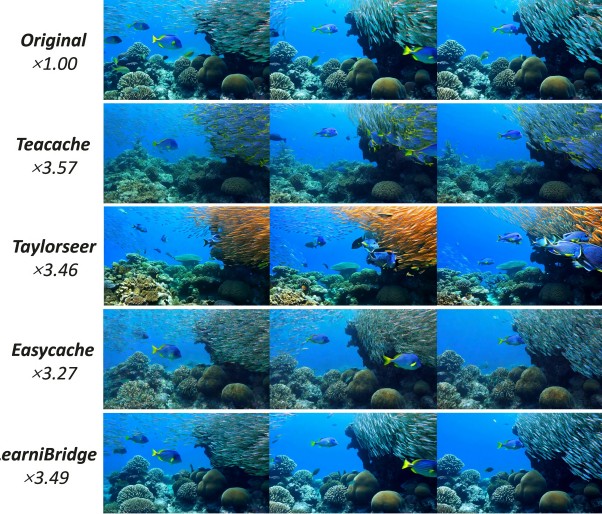

*Figure 6.* Visualization of different acceleration methods on WAN 2.1-1.3B. Baseline methods exhibit inconsistent color reproduction, degraded motion quality, and visible blurring, while *LearniBridge* maintains high visual quality close to the original video.

levels. Even under more aggressive acceleration at $5.75\times$, *LearniBridge* consistently maintains higher VBench scores and better perceptual metrics than TaylorSeer.

As shown in Table 3, we evaluate *LearniBridge* on the WAN 2.1-1.3B model and compare it with EasyCache, Tea-Cache, and TaylorSeer. Existing approaches are generally limited to approximately $3\times$ speedup and incur substantial degradation in output quality. In contrast, *LearniBridge* achieves speedup factors of $3.49\times$ (VBench 81.21) and $4.10\times$ (VBench 80.51), while consistently outperforming all competing methods across all quality metrics. Notably, at a $3.49\times$ speedup, *LearniBridge* preserves strong visual fidelity, achieving PSNR ($21.26 \uparrow$), SSIM ($0.8240 \uparrow$), and LPIPS ($0.1824 \downarrow$).

**Qualitative Study.** Figure 5 and Figure 6 present qualitative visual comparisons between *LearniBridge* and representative baseline methods on HunyuanVideo and WAN 2.1-1.3B, respectively. As illustrated in Figure 5, TaylorSeer and ToCa often introduce severe content inconsistency across frames. For example, the appearance of the fox, the shape of the strawberries, and the houses in snowy scenes undergo significant and unrealistic changes over time. TeaCache mainly suffers from degraded temporal dynamics, leading to unsmooth and unnatural motion. In Figure 5, the fox exhibits abnormal motion artifacts, such as the emergence of an extra leg. Similarly, in Figure 6, the school-of-fish scene becomes fragmented, resulting in noticeably discontinuous motion. EasyCache shows evident visual instability when the acceleration ratio exceeds $3\times$, manifesting as frame-wise flickering and floating artifacts that significantly deteriorate visual quality. In contrast, *LearniBridge* consistently preserves high fidelity to the original video content while maintaining strong temporal coherence, producing smoother, more stable, and visually consistent dynamic video generation results.

*Table 3.* Quantitative comparison in text-to-video generation for WAN 2.1-1.3B on VBench.

| Method | Acceleration | | | | Quality Metrics | | | |
|---|---|---|---|---|---|---|---|---|
| WAN 2.1-1.3B | Latency(s)↓ | Speed↑ | FLOPs(T)↓ | Speed↑ | VBench(%) | PSNR↑ | SSIM↑ | LPIPS↓ |
| Original | 291.55 | 1.00× | 13996.0 | 1.00× | 81.52 | – | – | – |
| EasyCache ($\delta$=0.13) (Zhou et al., 2025) | 89.16 | 3.27× | 4203.0 | 3.33× | 79.61 | 13.77 | 0.4745 | 0.4417 |
| TaylorSeer ($N$=4, $O$=2) (Liu et al., 2025b) | 84.26 | 3.46× | 4760.5 | 2.94× | 79.17 | 14.84 | 0.4456 | 0.4425 |
| TeaCache ($\delta$=0.2) (Liu et al., 2025a) | 81.67 | 3.57× | 3898.6 | 3.59× | 80.04 | 18.24 | 0.6804 | 0.3002 |
| **LearniBridge** ($N$=4) | 83.54 | 3.49× | 3802.3 | 3.68× | **81.21** | **21.26** | **0.8240** | **0.1824** |
| TeaCache ($\delta$=0.3) (Liu et al., 2025a) | 73.25 | 3.98× | 3364.4 | 4.16× | 79.23 | 12.82 | 0.4691 | 0.4327 |
| **LearniBridge** ($N$=5) | 71.11 | 4.10× | 3180.9 | 4.40× | **80.51** | **16.32** | **0.6254** | **0.3228** |

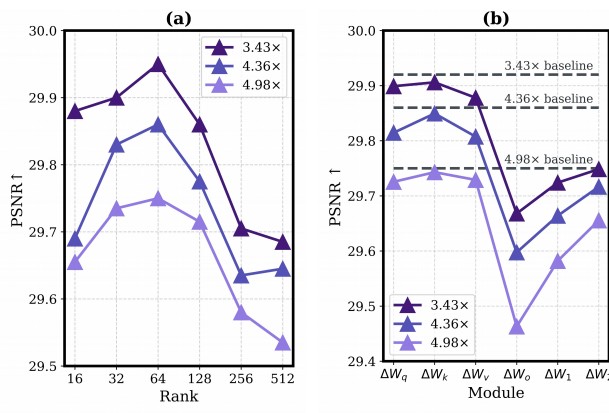

*Figure 7.* **(a)** Impact of varying the rank of LoRA adapters. As the rank increases, reconstruction quality first improves and then degrades, indicating that larger ranks do not necessarily lead to better calibration. **(b)** Impact of selectively removing LoRA adapters from different linear layers. All modules contribute to preserving high-fidelity reconstruction after acceleration.

*Table 4.* Ablation on calibration prompt length.

| Prompt Length | PSNR↑ | SSIM↑ | LPIPS↓ |
|---|---|---|---|
| Short (0–5 words) | 29.6247 | 0.6927 | 0.3135 |
| Medium (10–15 words) | **30.1569** | **0.7898** | 0.2654 |
| Long (20–25 words) | 30.0137 | 0.7878 | **0.2594** |

*Table 5.* Ablation on the number of calibration prompts.

| Number of Prompts | PSNR↑ | SSIM↑ | LPIPS↓ |
|---|---|---|---|
| 1 | 27.7913 | 0.6771 | 0.3829 |
| 3 | 29.1079 | 0.6970 | 0.3184 |
| 5 | 30.1623 | **0.7799** | 0.2681 |
| 10 | 30.0728 | 0.7765 | **0.2635** |
| 20 | **30.2297** | 0.7679 | 0.2746 |
| 30 | 30.0814 | 0.7642 | 0.2699 |

## 5. Ablation Studies

**Impact of Varying Rank** In this section, we evaluate the impact of varying the rank of the LoRA adapters on acceleration performance, with detailed results presented in Figure 7(a). We observe that the reconstruction quality, measured by PSNR, improves steadily as the rank increases, reaching a peak at $r = 64$. Beyond this point, further increasing the rank yields diminishing returns, indicating that a rank of 64 provides sufficient capacity to model the feature discrepancy between timesteps. This observation confirms that the calibration task can be effectively solved with a moderate parameter budget.

**Impact of Varying Module** In the experiments presented in Section 4, LoRA adapters are applied to all linear layers, including the query, key, value, and output projections, as well as the feed-forward layers, corresponding to $\Delta W_q$, $\Delta W_k$, $\Delta W_v$, $\Delta W_o$, $\Delta W_1$, and $\Delta W_2$. We investigate the impact of selectively removing individual LoRA adapters under different acceleration ratios to analyze how exclud-

ing a specific linear layer affects overall performance, as illustraed in Figure 7(b). Reconstruction quality is quantitatively evaluated using PSNR. Experimental results show that removing the LoRA adapters associated with the query, key, and value projection layers leads to a slight degradation in similarity to the original images. In contrast, excluding the adapters corresponding to the output projection ($W_o$) and the feed-forward network layers ($W_1$ and $W_2$) results in a more pronounced performance drop. This indicates that the dense feature transformations are more critical for calibration than the attention routing components.

**Impact of Calibration Prompts** We further study the influence of calibration prompts from two aspects: prompt length and prompt number. As shown in Table 4, medium and long prompts consistently outperform short prompts, with short prompts yielding notably lower SSIM and higher LPIPS. This indicates that overly simple inputs provide insufficient information for learning an effective calibration. Table 5 reports the effect of varying the number of calibration prompts. Performance improves substantially when increasing the number of prompts from 1 to 5, suggesting that a small prompt set is sufficient for calibration. Beyond 5 prompts, the performance largely saturates and only exhibits minor fluctuations.

# 6. Conclusion

In this paper, we present *LearniBridge*, a LoRA-based calibration method for cached features that accelerates diffusion models. We addressed the computational inefficiencies of Diffusion Transformers (DiTs) by investigating the nature of feature correction in acceleration tasks. Our analysis revealed that the requisite updates for feature reuse are characterized by a shared low-rank subspace across diverse prompts. Building on this, we proposed *LearniBridge*, a LoRA-based method that effectively calibrates historical features to bridge timestep discrepancies. Extensive experiments demonstrate that *LearniBridge* achieves up to $5.87\times$, $5.75\times$, and $4.10\times$ acceleration on FLUX, HunyuanVideo, and WAN 2.1, respectively, while preserving high-quality generation capabilities.

# Acknowledgment

This work is supported by the National Natural Science Foundation of China under Grant 62388102, by the Shenzhen Ubiquitous Data Enabling Key Lab under Grant ZDSYS20220527171406015, and by the Tsinghua Shenzhen International Graduate School-Shenzhen Pengrui Endowed Professorship Scheme of Shenzhen Pengrui Foundation.

# Impact Statement

This paper presents work whose goal is to advance the field of machine learning. There are many potential societal consequences of our work, none of which we feel must be specifically highlighted here.

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

# A. Detailed Experimental Settings

**Generation Resolution.** For text-to-image generation, FLUX.1-dev produces images at a resolution of $1024 \times 1024$. In the text-to-video domain, HunyuanVideo generates videos at a resolution of $544 \times 960 \times 49$ (height $\times$ width $\times$ frames), while WAN 2.1-1.3B generates videos at $832 \times 480 \times 81$.

**Prompt Set for LoRA Calibration.** As detailed in Table 6, we employ a compact set of 3–5 prompts for each model (FLUX.1-dev, HunyuanVideo, and WAN 2.1-1.3B). Our calibration prompts are randomly sampled from PartiPrompts. To increase semantic richness, these prompts are further extended in length using GPT-5.2. Rather than serving merely as independent static samples, these prompts act as seeds for generating full diffusion trajectories. During the pre-calibration pass, we extract extensive cross-timestep feature pairs from multiple steps along each trajectory, effectively amplifying the supervision signal.

**LoRA Training Configuration and Cost.** Detailed hyperparameters and resource metrics are provided in Table 7. Using the Adam optimizer, we observe that even with extended training epochs, the computational burden remains exceptionally low due to the parameter efficiency of our approach. Training requires only 0.33, 1.76, and 2.28 GPU·hours for FLUX.1-dev, HunyuanVideo, and WAN 2.1-1.3B, respectively. Peak memory consumption scales with model resolution, ranging from 10.67 GB to 49.05 GB. Crucially, the final LoRA module is lightweight ($< 0.6$ GB), ensuring that *LearniBridge* introduces negligible overhead in terms of both training compute and storage.

*Table 6.* Prompt examples used for training phase.

| FLUX.1-dev | HunyuanVideo | WAN 2.1-1.3B |
|---|---|---|
| A child chasing butterflies in a golden wheat field. | A hot pizza cools on a stone countertop. | A fox rotates its head left rapidly, sunlight glinting off its fur. |
| Subway train rushing past as commuters blur in motion. | | |
| Portrait painted in Cubist Picasso style. | A cat walks on the grass. | Blue fish with bright yellow fins weave through layered coral structures. |
| Ink wash painting of mountain and river, calm and balanced. | | |
| Lit alley in Tokyo with rain reflections. | A child runs across a grassy field. | A time-lapse of a peaceful lake under changing sky colors over a forest. |

*Table 7.* Training hyperparameters, computational cost, and LoRA parameter size.

| Model | Learning Rate | Epochs | Training Time (GPU·h) | Training Memory (GB) | LoRA Params (GB) |
|---|---|---|---|---|---|
| FLUX.1-dev | $1.0 \times 10^{-3}$ | 500 | 0.33 | 10.67 | 0.575 |
| HunyuanVideo | $1.5 \times 10^{-3}$ | 700 | 1.76 | 49.05 | 0.588 |
| WAN 2.1-1.3B | $1.0 \times 10^{-4}$ | 600 | 2.28 | 23.99 | 0.192 |

# B. Additional Empirical Analysis

In this section, we provide extended experimental results to further validate the intrinsic properties of the feature calibration update discussed in Section 3.2. Specifically, we examine the consistency of the low-rank structure across different diffusion timesteps and verify the universality of the update direction across all linear layers of the model.

## B.1. Temporal Consistency of the Intrinsic Low-Rank Structure

In the Figure 1, we demonstrated the rapid decay of singular values for the input matrix $X_t^l$ at a representative timestep $t = 7$. Here, we extend this analysis to the entire diffusion process. Figure 8 illustrates the singular value spectrum at distinct timesteps $t \in \{17, 27, 37, 47\}$. We observe that the spectral energy remains concentrated in the top principal components regardless of the diffusion stage. This confirms that the intrinsic low-rank structure is a temporally consistent property, strictly justifying our use of a low-rank adapter throughout the generation process.

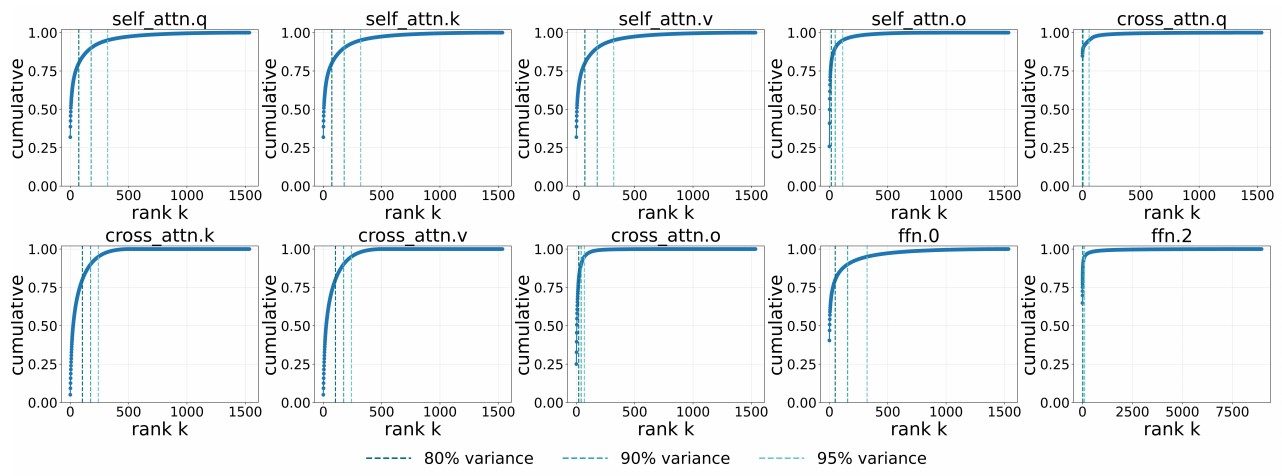

(a) $t = 17$

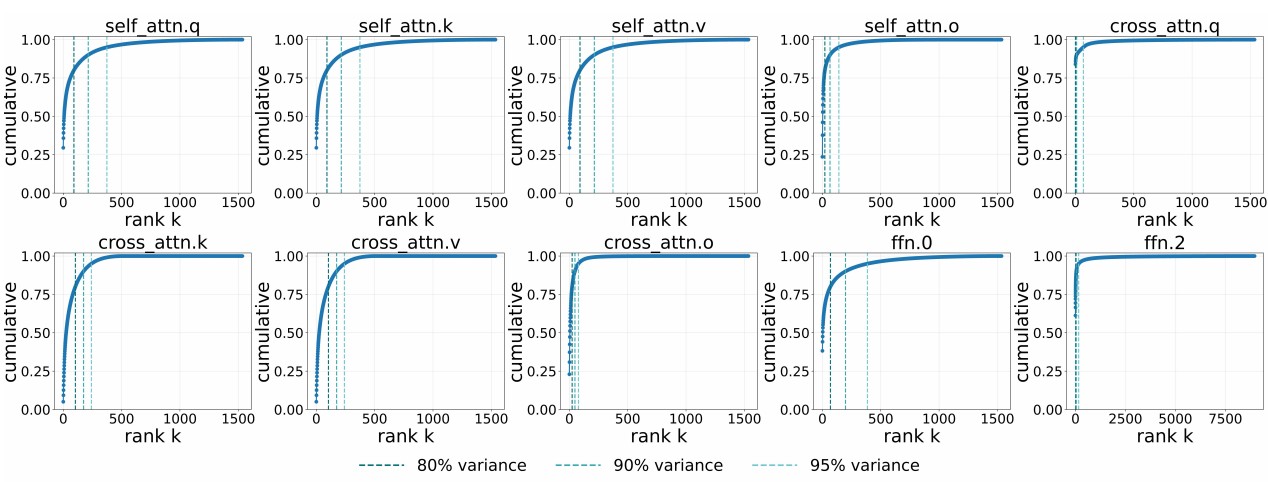

(b) $t = 27$

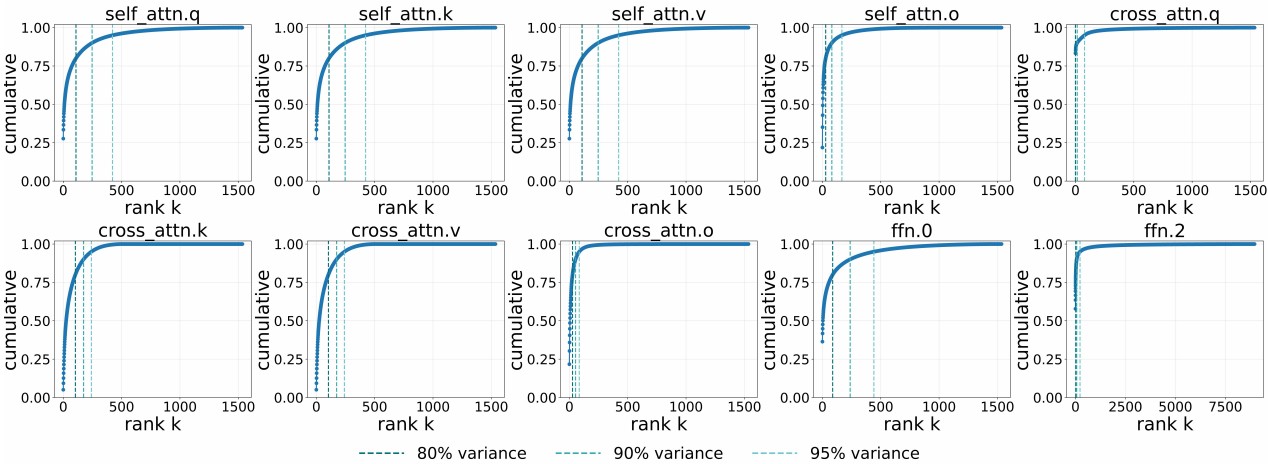

(c) $t = 37$

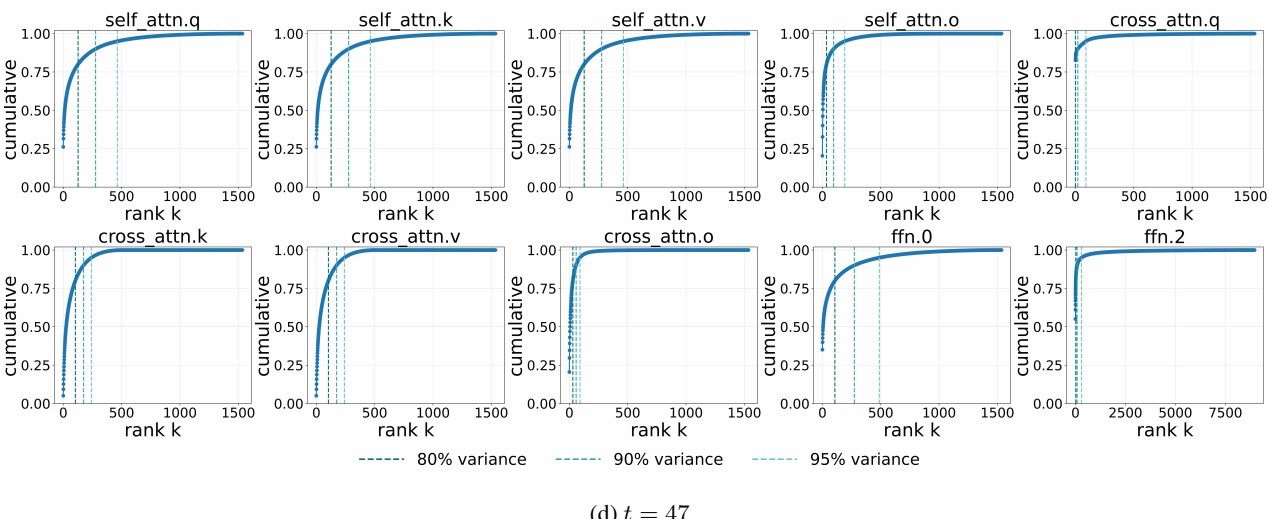

(d) $t = 47$

*Figure 8.* We visualize the SVD analysis of the aggregated input matrix $X_t^l$ at four distinct diffusion timesteps ($t \in \{17, 27, 37, 47\}$). Across all stages, the singular values exhibit a consistent rapid decay, indicating that the spectral energy is invariably concentrated in the top principal components. This validates that the low-rank constraint on $\Delta W^l$ remains effective throughout the temporal progression of diffusion.

## B.2. Layer-Wise Universality of the Prompt-Invariant Calibration Direction

To demonstrate that the prompt-invariance of the calibration update is not limited to the query projection layer of the cross-attention modules analyzed in Figure. 2, we further evaluate the subspace similarity across all linear layers in the model. Figure 10 presents the principal angles between updates computed from disjoint prompt groups for every linear layer. The results show consistently small angles globally, reinforcing the conclusion that the calibration direction follows a universal pattern independent of the specific input prompts and network depth.

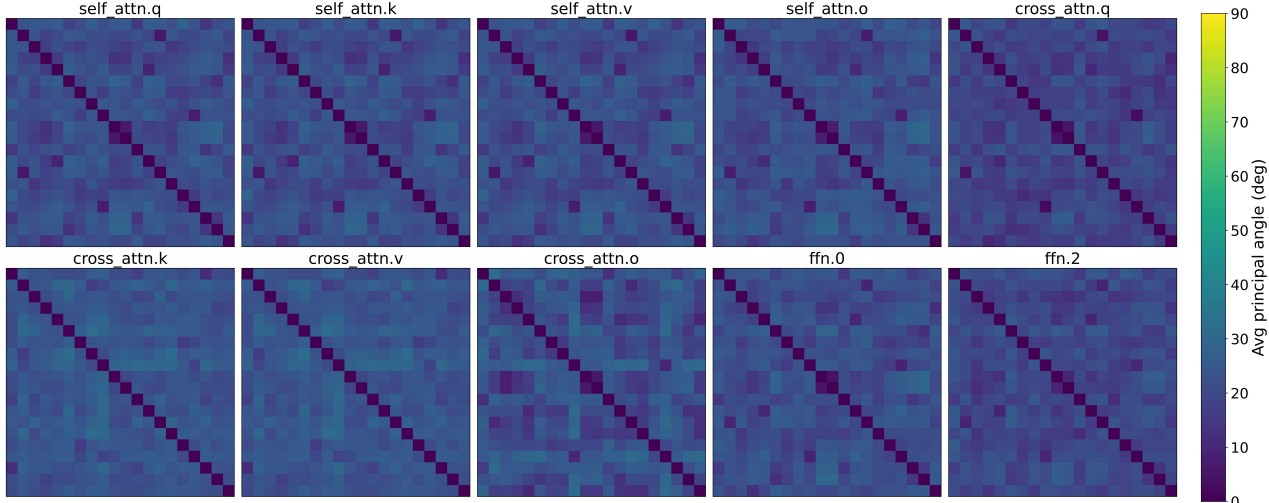

*Figure 9.* Global prompt-shared subspace analysis across all linear layers. The heatmap visualizes the pairwise subspace angles between updates derived from disjoint prompt groups. The angles remain consistently low across the entire network architecture, confirming the global universality of the learned calibration direction.

# C. More Results for LEARNIBRIDGE

In this section, we provide additional qualitative results for text-to-image and text-to-video generation tasks. Figure. 10 presents qualitative results for text-to-image generation on FLUX.1-dev. Additional qualitative results for text-to-video generation on HunyuanVideo and WAN 2.1-1.3B are provided in the supplementary material. Consistent with the observations in Section 4.2, *LearniBridge* preserves visual quality while enabling significant inference acceleration, further validating its effectiveness.

## C.1. Text-to-Image Generation

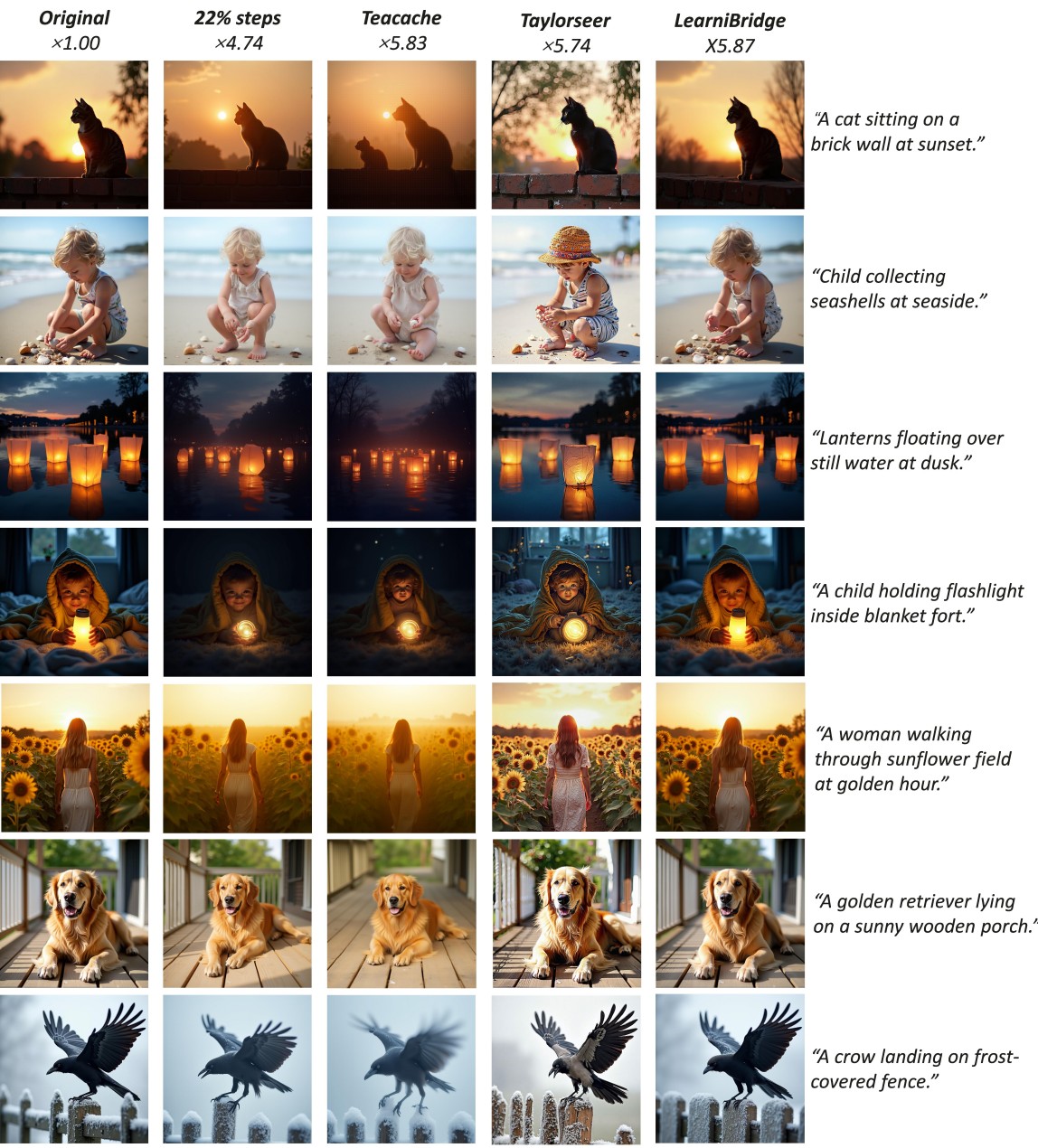

*Figure 10.* More qualitative comparisons of text-to-image generation on FLUX.1-dev.

## C.2. Text-to-Video Generation

Qualitative results for text-to-video generation on HunyuanVideo and WAN 2.1-1.3B are provided in the supplementary material. The corresponding text prompts used in our experiments are listed in Table 8.

*Table 8.* Text prompts used for text-to-video generation.

| Model | Text Prompt |
|---|---|
| HunyuanVideo | Snow falls gently on a quiet village. 
 A bowl of strawberries rests on a linen cloth. 
 A dog naps beside a crackling fireplace. 
 A fox trots across a sunlit forest path. |
| WAN 2.1-1.3B | A vibrant underwater scene. A group of blue fish, with yellow fins, are swimming around a coral reef. The coral reef is a mix of brown and green, providing a natural habitat for the fish. The water is a deep blue, indicating a depth of around 30 feet. The fish are swimming in a circular pattern around the coral reef, indicating a sense of motion and activity. The overall scene is a beautiful representation of marine life. 
 A cute and happy Corgi playing joyfully in a picturesque park during a beautiful sunset. The Corgi has fluffy white fur with a playful expression, wagging its tail excitedly as it runs around. The dog bounces along the grass, occasionally sniffing at flowers and chasing after butterflies. The park is filled with lush greenery, colorful wildflowers, and vibrant foliage. A warm golden hue blankets the sky, casting a serene and peaceful atmosphere. The scene transitions from a wide angle to a close-up shot focusing on the Corgi's energetic play. 
 A cat and a dog baking a cake together in a kitchen. The cat is carefully measuring flour, while the dog is stirring the batter with a wooden spoon. The kitchen is cozy, with sunlight streaming through the window. 
 A leisurely boat sailing along the picturesque Seine River, with the Eiffel Tower towering in the misty background. The camera captures the calm waters, ripples, and reflections in super slow motion. Passengers appear content and at peace as the scene transitions into a dreamlike night sky. |

