# OpenReview forum: "LearniBridge: Learnable Calibration of Feature Caching for Diffusion Models Acceleration"
_ICML.cc/2026/Conference — ICML 2026 regular_

### Official Review · Reviewer_mtax · 2026-03-08

**Soundness:** 3
**Presentation:** 3
**Significance:** 3
**Originality:** 3
**Overall Recommendation:** 4
**Confidence:** 3

**Summary:**

This paper introduces "LearniBridge," a framework designed to accelerate Diffusion Transformers (DiTs) via a learnable calibration mechanism for feature caching. The proposed  approach is based on the insight that feature shifts across diffusion timesteps reside within a prompt-invariant, low-dimensional subspace. Leveraging this property, the method employs lightweight LoRA adapters to model the temporal evolution of features. In contrast to existing heuristic-based methods like TaylorSeer and DeepCache, LearniBridge utilizes a data-driven approach. By injecting LoRA updates exclusively into the final Transformer block, the framework aims to reconstruct accurate representations for skipped timesteps with minimal computational overhead. The authors validate the performance of their method on public benchmarks, by comparing with state-of-the-art approaches.

**Compliance With Llm Reviewing Policy:**

Affirmed.

**Final Justification:**

The rebuttal removes most of my concerns and I maintain my score.

**Key Questions For Authors:**

1. The experiments are currently conducted exclusively on the Ascend 910B platform. Given that low-level operator optimizations vary significantly across different hardware architectures, this could lead to differing acceleration ratios or even impact the final generation quality. While I understand that running a full suite of evaluations on a new hardware architecture during the short rebuttal period is challenging, could the authors provide some preliminary results on a widely adopted NVIDIA GPU architecture (e.g., Hopper or Ampere) for at least one representative model? Alternatively, if this is not feasible, I recommend explicitly discussing this hardware specificity as a limitation and outlining the expected generalization behaviors in the revised manuscript.

2. While Figure 2 illustrates prompt-invariance through the analysis of small principal angles, this alone is insufficient to conclusively prove strict prompt-invariance. I suggest including ablation studies to further validate this claim. For instance, evaluating the method's performance using prompts that are entirely different from the calibration set (out-of-domain), as well as highly similar prompts, would better substantiate the robustness and consistency of the approach across varying prompt distributions.

3. This paper currently does not report the theoretical savings in computational complexity (e.g., FLOPs reduction). Although empirical latency is crucial, FLOPs provides a hardware-agnostic metric that helps decouple algorithmic efficiency from low-level hardware optimizations. Since the compared baseline methods provide this data, its absence makes it difficult to fully contextualize the theoretical advantages of LearniBridge. Could the authors provide a comparison of the theoretical computational reduction to complement the empirical speedups? This addition would significantly strengthen the evaluation and help address my concerns regarding cross-platform performance.

4. The authors critique TaylorSeer by stating it "requires more cached features, limiting its use for long video sequences." However, the experimental section lacks a rigorous comparison on truly long video sequences to substantiate this claim. The video lengths generated by the evaluated models do not appear to fall into the typical "long sequence" category. The authors should either provide comparative experiments on extended video generation tasks to support their claim or tone down the assertions regarding their method's superiority in handling long sequences.

**Limitations:**

No. The authors have not adequately discussed the limitations of their work, and I suggest they expand this section to address the following points: The proposed approach is training/calibration-based. Compared to existing training-free baseline methods, it inherently lacks full "plug-and-play" flexibility. Furthermore, if users wish to achieve different acceleration ratios, they cannot simply tune hyperparameters dynamically during inference, as is typical with training-free methods. This requirement inevitably introduces higher deployment complexity and additional usage costs in practical scenarios. I strongly encourage the authors to be upfront about this limitation and explicitly discuss the trade-off between the method's performance gains and its operational flexibility.

**Strengths And Weaknesses:**

Strengths:
+The paper provides a valuable insight by uncovering the prompt-invariant, low-dimensional subspace structure of feature shifts in diffusion trajectories. By shifting the paradigm from heuristic-based "prediction" (like TaylorSeer) to data-driven "calibration," it offers a more robust solution for error accumulation. The discovery that only 3-5 samples are needed for global generalization is highly significant for practical deployment.
+The theoretical foundation is solid, supported by rigorous SVD analysis that proves the concentrated spectral energy of feature residuals across different timesteps and prompts. The experimental evaluation is comprehensive thorough, testing on the latest SOTA models like FLUX.1-dev, HunyuanVideo, and WAN 2.1. The ablation studies on LoRA rank and module sensitivity effectively justify the "calibrate final block" design.

Weaknesses:
-First, the paper lacks an analysis of theoretical computational complexity (FLOPs). Given that prior baseline methods provide this metric, its absence makes it difficult to definitively ascertain the theoretical computational advantages of the proposed method. Second, while the introduction claims that existing methods struggle with supporting long sequences, the experimental section does not include comparative evaluations on long video generation tasks to substantiate any significant improvements in this area.Then,all latency and throughput benchmarks are reported on the Ascend 910B platform. While valuable for showcasing hardware diversity, adding comparisons on standard NVIDIA Hopper/Blackwell architectures would make the efficiency gains more universally interpretable for the global research community.
-Although the training cost is minimal (~2 GPU hours), the method still requires a pre-calibration phase, making it slightly less "plug-and-play" than purely training-free methods like TeaCache or TaylorSeer. This introduces a small but non-zero barrier for users who require immediate, zero-shot acceleration across rapidly changing model checkpoints.

---

> ### Author Rebuttal · Authors · 2026-03-31
>
> We thank you for recognizing the insights of our low-dimensional subspace discovery, the data efficiency of our method, and the solid theoretical and experimental foundation of our work. We appreciate your constructive feedback and address your detailed questions below:
>
> > **[Q1] Hardware specificity (Ascend 910B vs. NVIDIA GPUs).**
>
> Due to laboratory hardware constraints, running a full suite of evaluations on NVIDIA Hopper/Ampere architectures is unfortunately unfeasible.
> However, we emphasize that LearniBridge is a strictly hardware-decoupled approach, as it fundamentally reduces the computational overhead within the DiT architecture itself. The significant reduction in theoretical FLOPs (presented below) serves as direct evidence of our method's efficiency.
>
> > **[Q2] Insufficient evidence for strict prompt-invariance and lack of OOD evaluation.**
>
> * The 100 calibration prompts used in Figure 2 were randomly sampled from *PartiPrompts*, naturally encompassing a wide variance of semantic features. The observed small principal angles hold consistently across this diverse set.
> * We evaluated the calibrated module on a completely different benchmark: 200 prompts from *DrawBench*. DrawBench covers a highly diverse set of prompt categories, including **conflicting logic**, **misspellings**, **rare words**, and complex compositional constraints (**counting, positional, colors, and long descriptions**). The consistent performance across these entirely different categories demonstrates the robustness of our approach.
>
> > **[Q3] Computational complexity (FLOPs) reduction.**
>
> As shown in the table below, We have measured the FLOPs reduction for all our evaluated models.
>
> **FLUX Efficiency Comparison**
>
> | Method | Latency (s) ↓ | Speed ↑ | FLOPs (T) ↓ | Speed ↑ |
> | :--- | :---: | :---: | :---: | :---: |
> | Original | 27.32 | 1.00× | 3719.5 | 1.00× |
> | 22% steps | 6.00 | 4.55× | 817.5 | 4.55× |
> | ToCA (N=9) | 6.88 | 3.97× | 854.4 | 4.35× |
> | TeaCache (δ=0.8) | 6.60 | 4.14× | 892.0 | 4.17× |
> | TaylorSeer (N=5, O=2) | 6.20 | 4.41× | 817.5 | 4.55× |
> | **LearniBridge (N=5)** | 6.27 | 4.36× | 839.6 | 4.43× |
> | ToCA (N=10) | 5.78 | 4.73× | 714.7 | 5.20× |
> | TeaCache (δ=1.0) | 5.66 | 4.83× | 743.6 | 4.89× |
> | TaylorSeer (N=6, O=2) | 5.47 | 4.99× | 745.4 | 4.99× |
> | **LearniBridge (N=6)** | 5.61 | 4.87× | 759.1 | 4.90× |
> | ToCA (N=12) | 4.65 | 5.87× | 628.3 | 5.92× |
> | TeaCache (δ=1.4) | 4.56 | 5.99× | 603.8 | 6.16× |
> | TaylorSeer (N=8, O=2) | 4.48 | 6.10× | 596.1 | 6.24× |
> | **LearniBridge (N=8)** | 4.43 | 6.17× | 599.9 | 6.20× |
>
> **HunyuanVideo Efficiency Comparison**
>
> | Method | Latency (s) ↓ | Speed ↑ | FLOPs (T) ↓ | Speed ↑ |
> | :--- | :---: | :---: | :---: | :---: |
> | Original | 617.79 | 1.00× | 29773.0 | 1.00× |
> | 22% steps | 135.78 | 4.55× | 6550.1 | 4.55× |
> | Teacache (δ=0.3) | 167.88 | 3.68× | 7794.0 | 3.82× |
> | TaylorSeer (N=4, O=1) | 161.30 | 3.83× | 7733.2 | 3.85× |
> | LearniBridge (N=4) | 164.74 | 3.75× | 7876.5 | 3.78× |
> | ToCa (N=5) | 149.94 | 4.12× | 7005.4 | 4.25× |
> | Teacache (δ=0.4) | 130.89 | 4.72× | 6151.5 | 4.84× |
> | TaylorSeer (N=5, O=1) | 132.86 | 4.65× | 5966.5 | 4.99× |
> | **LearniBridge (N=5)** | 125.31 | 4.93× | 5966.5 | 4.99× |
> | TaylorSeer (N=7, O=1) | 103.30 | 5.98× | 4771.3 | 6.24× |
> | **LearniBridge (N=7)** | 100.45 | 6.15× | 4794.4 | 6.21× |
>
> **Wan2.1 Efficiency Comparison**
>
> | Method | Latency (s) ↓ | Speed ↑ | FLOPs (T) ↓ | Speed ↑ |
> | :--- | :---: | :---: | :---: | :---: |
> | Original | 291.55 | 1.00× | 13996.0 | 1.00× |
> | EasyCache (δ=0.13) | 89.16 | 3.27× | 4203.0 | 3.33× |
> | TaylorSeer (N=4, O=2) | 84.26 | 3.46× | 3920.4 | 3.57× |
> | TeaCache (δ=0.2) | 81.67 | 3.57× | 3898.6 | 3.59× |
> | **LearniBridge (N=4)** | 83.54 | 3.49× | 3802.3 | 3.68× |
> | TeaCache (δ=0.3) | 73.25 | 3.98× | 3364.4 | 4.16× |
> | **LearniBridge (N=5)** | 71.11 | 4.10× | 3180.9 | 4.40× |
>
> > **[Q4] Claims regarding long sequence generation compared to TaylorSeer.**
>
> We clarify that our original statement ("*...limiting its use for long video sequences...*") was motivated by memory footprint limitations. For instance, when generating $544 \times 960 \times 65$ videos on HunyuanVideo, caching multiple historical features (as required by TaylorSeer) leads to Out-Of-Memory  errors, whereas LearniBridge succeeds.
> However, we agree this claim falls outside our primary focus. We have therefore removed it from the revised manuscript.
>
> > **Limitations regarding training/calibration and plug-and-play flexibility.**
>
> We agree that the phrase "*...yielding a plug-and-play calibration module*" in our original manuscript is imprecise. While LearniBridge is highly data-efficient, it still requires a brief calibration phase and lacks the immediate, zero-shot flexibility of entirely training-free methods.
> We have removed this wording to avoid overstatement and explicitly added this operational trade-off to the limitations section in the revised manuscript.

---

> > ### Author Rebuttal · Reviewer_mtax · 2026-04-04
> >
> > Thanks for the authors' response. I will keep my initial rating.

---

> > > ### Author Response · Authors · 2026-04-08
> > >
> > > We sincerely thank the reviewer for their time, valuable feedback, and continued engagement with our work throughout the review process. We are very glad that our responses have addressed your concerns, and we deeply appreciate your final support for our manuscript.

---

### Official Review · Reviewer_z8gt · 2026-03-11

**Soundness:** 3
**Presentation:** 3
**Significance:** 3
**Originality:** 2
**Overall Recommendation:** 5
**Confidence:** 4

**Summary:**

The paper introduces LearniBridge, a method designed to significantly accelerate Diffusion Transformers for image and video generation while maintaining quality. As we know, DiTs have achieved strong results in generative tasks, but are computationally expensive.    Many recent papers developed acceleration techniques relying on feature caching, which reuses intermediate representations from earlier diffusion steps. But these approaches often suffer from error accumulation, especially at high acceleration levels. The authors analyse this problem and show that the optimal correction for cached features lies in a shared low-rank subspace across different prompts, meaning the necessary adjustments can be efficiently modeled with low-dimensional updates. Based on this, the authors propose a lightweight calibration mechanism using LoRA-based updates that bridges features across diffusion timesteps and corrects errors introduced by caching requireing only 3–5 training samples to learn the calibration. Experiments on image and video generation models show substantial inference speedups of up to 5.87× on FLUX, 5.75× on HunyuanVideo, and 4.10× on WAN 2.1 while preserving or improving quality.

**Compliance With Llm Reviewing Policy:**

Affirmed.

**Final Justification:**

I am increasing my score to 5 as the rebuttal addresses my concerns and queries.

**Key Questions For Authors:**

The paper abstract says "Our code is included in the supplementary material and will be released on GitHub." But I could not find any code related info in the supp material.

The method assumes the correction can be represented as linear. But diffusion feature dynamics are highly nonlinear across timesteps. How the method will handle non linearity. Does using small nonlinear residual network instead of LORA make sense?

**Limitations:**

no limitation and impact statement being included. can this work be used to accelerate deepfake video/image generation even on smaller GPUs?

**Strengths And Weaknesses:**

Strengths-  The idea is quite interesting of doing calibration of features to accelerate text to image and video inference.  Diffusion models require evaluating the backbone network at every timestep, making inference expensive. Feature caching reduces this cost by reusing intermediate representations rather than recomputing them.This is an important area of research directly addressing efficient/sustainable AI.

The key contribution is the observation that cross-timestep feature corrections lie in a low-rank subspace. This provides a principled justification for using LoRA rather than a totally heuristic approximation.

The method is extremely data efficient as it only requires a few samples for calibration.

The results on text to video generation for Huanyuan video is better than baselines atleast for SSIM and LPIP metrics.

The paper is well written and structured.

Good ablation studies.

Weakness-

The correction is only applied to final block. It might be worth while to try correction at more blocks to see if it helps improve the performance. I understand that this might lead to decreased acceleration.

The low-rank observation is empirical and somewhat justified. However the authors do not provide any theoretical guarantees on sample  quality or approximation error.

The results are marginally better with respect to baselines. In case of text to image on Drawbench the results are marginally better.

The qualitative results holds very less value as good examples can always be cherry picked. A more rigorous benchmark is required to validate such results.

Paper saying code is in supp material but I could not find it.

No comparison with Consistency Models or Progressive Distillation based models. They can do inference in a few steps.

The paper is using Flux as a base model. It will be worth to try other model such as Luminal-Image-2.0 to show that the proposed approach works well across other models.

Overall the idea is good yet incremental. It brings in calibration of feature caching. This is similar to calibration done during the quantisation process finetuning stage to recover lost accuracy.

---

> ### Author Rebuttal · Authors · 2026-03-31
>
> We thank you for recognizing the effectiveness of our method, the insight of our low-rank calibration, and the data efficiency of our approach. We appreciate your constructive feedback, which helps us further improve the paper. We address your questions and clarify the key points below:
>
> > **[W1] Impact of applying the correction to more blocks.**
>
> We conducted additional ablation studies to apply the correction mechanism to multiple blocks (specifically, the last 1, 2, and 3 blocks) on FLUX. we generate 200 images using prompts from the DrawBench benchmark with a skip interval of $N = 5$. We evaluate PSNR, SSIM, and LPIPS, alongside the additional FLOPs incurred.
>
> | Blocks Corrected | PSNR ↑ | SSIM ↑ | LPIPS ↓ | Additional FLOPs (T) |
> | :--- | :---: | :---: | :---: | :---: |
> | Last 1 Block (Ours) | 30.1242 | 0.7880 | 0.2634 | 0 |
> | Last 2 Blocks | 30.1623 | 0.7798 | 0.2754 | +12.39 |
> | Last 3 Blocks | 30.0137 | 0.7868 | 0.2594 | +24.80 |
>
> Since correcting additional blocks linearly increases computational overhead without yielding consistent quality gains, we apply the correction solely to the final block. Please see our response to **Reviewer mtax (Q1)** for detailed FLOPs counts.
>
> > **[W6]  Comparison with Consistency/Distillation Models.**
>
> Our work focuses on lightweight, calibration-based acceleration rather than the high-cost distillation paradigm. Distillation requires massive training resources to alter the model's trajectory, which is fundamentally different from our few-shot approach. We therefore leave the systematic comparison and potential integration with such models for future exploration.
>
> > **[W7]  Evaluation on other models.**
>
> Our evaluation deliberately selects FLUX, HunyuanVideo, and WAN 2.1, as they represent the most widely adopted and state-of-the-art architectures in recent diffusion acceleration literature [1][2][3][4]. Demonstrating effectiveness across these representative foundational models is the standard practice for validating acceleration mechanisms. We appreciate the suggestion and will leave the specific adaptation for Lumina-Image-2.0 and other emerging architectures for future exploration.
>
> > **[Q1] Clarification on the submitted code.**
>
> We sincerely apologize for any confusion caused during the review process. We have double-checked our submission and respectfully clarify that the source code, specifically the implementation for adapting FLUX, was indeed included in the uploaded zip file under the directory `supplementary_material/learnibridge_code`.
> We are fully committed to open source and will ensure the complete, cleaned-up codebase for all models (including FLUX, HunyuanVideo, and WAN 2.1) is publicly released upon publication.
>
> > **[Q2] Linear assumption versus nonlinear dynamics.**
>
> While diffusion dynamics are inherently non-linear, employing linear or polynomial approximations for feature compensation is a well-established and effective path in the acceleration literature. For instance, TaylorSeer [1] utilizes first-order Taylor expansion—fundamentally a linear approximation—to estimate caching errors.
> Building upon this simple and effective linear assumption, our method proposes an correction mechanism in the weight domain.
>
> **References:**
> [1] Liu J, Zou C, Lyu Y, et al. From reusing to forecasting: Accelerating diffusion models with taylorseers. *ICCV 2025.*
> [2] Son B, Jeon J, Choi J, et al. Relational Feature Caching for Accelerating Diffusion Transformers. *ICLR 2025.*
> [3] Bu J, Ling P, Zhou Y, et al. DiCache: Let Diffusion Model Determine Its Own Cache. *ICLR 2025.*
> [4] Feng L, Zheng S, Liu J, et al. Hicache: Training-free acceleration of diffusion models via hermite polynomial-based feature caching. *ICLR 2025.*

---

> > ### Author Rebuttal · Reviewer_z8gt · 2026-04-02
> >
> > Only some of my concerns are addressed. Mainly, the gains in improvement are tiny, and there is no comparison with other approaches, such as consistency-based approaches. The paper is using Flux as a base model. It will be worth trying other models, such as Luminal-Image-2.0, to show that the proposed approach works well across other models; cherry picked examples.

---

> > > ### Author Response · Authors · 2026-04-08
> > >
> > > We appreciate the reviewer's valuable time in reviewing our rebuttal and sharing further constructive feedback. Please find our detailed responses to your remaining concerns below.
> > >
> > > > **Q1: The gains in improvement are tiny.**
> > >
> > > LearniBridge provides robust SOTA improvements across leading models, achieving a +1.52dB PSNR on FLUX at 5.87x, a +1.08dB PSNR on HunyuanVideo at 3.75x, and a +1.28% VBench score on Wan2.1 at 4.1x, representing significant contributions under high acceleration ratios.
> > >
> > > > **Q2: There is no comparison with other approaches, such as consistency-based approaches.**
> > >
> > > To further demonstrate the versatility and efficiency of LearniBridge, we expanded our experiments on **FLUX.1-dev** to include comparisons with quantization (8-bit) and architecture-based distillation (Flux-mini). The results are summarized in the table below:
> > >
> > > | Method | PSNR $\uparrow$ | SSIM $\uparrow$ | LPIPS $\downarrow$ | Acceleration |
> > > | :--- | :--- | :--- | :--- | :--- |
> > > | 8-bit Quantization | 31.3893 | 0.8171 | 0.1859 | $2.10\times$ |
> > > | **LearniBridge ($N=2$)** | **33.1893** | **0.8471** | **0.1059** | $1.72\times$ |
> > > | **8-bit + LearniBridge ($N=2$)** | **32.6214** | **0.8126** | **0.1404** | $3.49\times$ |
> > > | Flux-mini (Distilled) | 27.2825 | 0.5084 | 0.4857 | $3.80\times$ |
> > > | **LearniBridge ($N=5$)** | **30.1525** | **0.7879** | **0.2682** | $4.36\times$ |
> > >
> > > **Key Observations:**
> > >
> > > * **Orthogonality with Quantization:** LearniBridge is orthogonal to quantization methods. By combining LearniBridge with an 8-bit quantized model, we achieve **3.49$\times$** acceleration.
> > > * **Competitiveness against Distillation:** While distillation-based approaches like Flux-mini can offer higher objective performance at high acceleration ratios, they inherently demand massive training resources. **LearniBridge ($N=4$)** achieves a remarkable **$4.36\times$** acceleration while maintaining strong perceptual fidelity, offering an excellent computation-quality trade-off.
> > >
> > > > **Q3: The paper is using Flux as a base model. It will be worth trying other models, such as Luminal-Image-2.0, to show that the proposed approach works well across other models.**
> > >
> > > We have evaluated the acceleration performance of **LearniBridge** on **Lumina-Image-2.0** and provided a comprehensive comparison with **TeaCache**. The results, as shown in the table below, consistently demonstrate that our approach maintains superior image quality and higher acceleration across different models, thereby confirming the effectiveness and generalizability of our method.
> > >
> > > | Method | PSNR $\uparrow$ | SSIM $\uparrow$ | LPIPS $\downarrow$ | Acceleration |
> > > | :--- | :--- | :--- | :--- | :--- |
> > > | TeaCache ($\delta=1$) | 28.8574 | 0.8342 | 0.1610 | $1.85\times$ |
> > > | **LearniBridge ($N=2$)** | **29.3798** | **0.8462** | **0.1547** | **$1.92\times$** |
> > > | TeaCache ($\delta=1.5$) | 28.3355 | 0.7689 | 0.2471 | $2.63\times$ |
> > > | **LearniBridge ($N=3$)** | **28.5017** | **0.7879** | **0.2346** | **$2.78\times$** |
> > > | TeaCache ($\delta=2$) | 26.5629 | 0.7439 | 0.3020 | $3.33\times$ |
> > > | **LearniBridge ($N=4$)** | **27.5743** | **0.7512** | **0.2846** | **$3.57\times$** |
> > >
> > > >**Q4: Cherry picked examples.**
> > >
> > > To ensure an objective evaluation and avoid selection bias, our main text relies on standard quantitative benchmarks rather than isolated visual examples. Following the common evaluation protocols established by prior diffusion acceleration works [1][2][3], we validate LearniBridge across large-scale datasets:
> > >
> > > * **Image Generation (DrawBench):** We evaluated all **200 test images** across five metrics: **ImageReward $\uparrow$**, **CLIPScore $\uparrow$**, **PSNR $\uparrow$**, **SSIM $\uparrow$**, and **LPIPS $\downarrow$**.
> > > * **Video Generation (VBench):** We evaluated our method on **4,730 video samples** using **VBench(%) $\uparrow$**, **PSNR $\uparrow$**, **SSIM $\uparrow$**, and **LPIPS $\downarrow$**.
> > >
> > > **Reference**
> > > [1] Son B, Jeon J, Choi J, et al. Relational Feature Caching for Accelerating Diffusion Transformers. *ICLR 2026.*
> > > [2] Liu Q, et al. From Reusing to Forecasting: Accelerating Diffusion Models with TaylorSeers. *ICCV 2025.*
> > > [3] Feng L, Zheng S, Liu J, et al. Hicache: Training-free acceleration of diffusion models via hermite polynomial-based feature caching. *ICLR 2026.*

---

### Official Review · Reviewer_CVzy · 2026-03-12

**Soundness:** 3
**Presentation:** 3
**Significance:** 2
**Originality:** 3
**Overall Recommendation:** 4
**Confidence:** 4

**Summary:**

The paper introduces LearniBridge, a learnable low-rank adapter for predicting the future features of DiT blocks in image/video generative models, demonstrating that these features exhibit a low-rank, prompt-invariant structure. Experiments on DrawBench for images and VBench for videos using FLUX, HunyuanVideo and Wan2.1 models show that it achieves upto 4x-5x speedup while maintaining the visual quality.

**Compliance With Llm Reviewing Policy:**

Affirmed.

**Final Justification:**

The rebuttal addressed most of my concerns and I have raised my score to 4.

**Key Questions For Authors:**

- See weaknesses above.

- What is the layer and timestep used in the analysis of Figure 1? What are the 100 prompts used in this analysis?

- How are the prompts used for pre-calibration chosen? Table 4 shows some examples of these prompts. Do they overlap with the prompts in the evaluation benchmarks? What are the prompts used for text-to-image generation (Table 1)?

- Why does training LoRA adapters requires 500-700 epochs as shown in Table 5 (Appendix)? LoRA updates typically require only few epochs of training.

**Limitations:**

The paper does not discuss the limitations of the method

**Strengths And Weaknesses:**

**Strengths**

- The paper is well-motivated and the method is easy to follow.

- The proposed method is shown to be invariant to prompts and requires only a small calibration set.

- It shows better visual consistency to the original outputs compared to prior methods.

**Weaknesses**

- The novelty of the proposed method is limited as prior method TaylorSeer already shows that the intermediate features of Diffusion transformers can be predicted based on features from previous timesteps. The proposed method extends this idea using LoRAs for prediction.

- CLIP-score metric reported in Table 1 (~0.8) is different from the standard (0.2-0.3) CLIP score range. How is it computed?

- The reported results in Tables 1 and 2 for prior methods are different from the paper (e.g., TaylorSeer). Why is there a discrepancy?

- Quantitative results are missing standard visual quality metrics such as FID/FVD and HPS v2 scores as well the FLOPs. How does it compare with prior methods?

- Performance improvements over prior methods such as TaylorSeer is limited as seen from Table 2 especially for large acceleration ratio (N=7). It shows that the gains diminish at higher acceleration ratios.

- LoRA adapter training still requires significant memory overhead as seen in Appendix A.

---

> ### Author Rebuttal · Authors · 2026-03-31
>
> Thanks for your positive feedback on our method's clarity and visual performance. We address your remaining questions below:
>
> > **[W1] Clarification on Novelty.**
>
> While predicting intermediate features from previous timesteps is the shared foundation of feature caching methods, contemporary works(ICLR 2025) primarily innovate by better utilizing these intermediate features for improved prediction or scheduling strategies [1][2][3]. **Our Innovation:** LearniBridge establishes the first paradigm of cache error calibration, utilizing LoRA updates to reconstruct future-step representations from cached features, achieving up to 5.87x acceleration. The novelty of our work has been generally acknowledged by the other reviewers.
>
> > **[W2] Different CLIPScore range.**
>
> The scale difference arises from two formulations below:
>
> **(1) Unscaled CLIPScore (Ours, ~0.8):** Directly computes the cosine similarity [4].
>
> $$\text{CLIPScore}_{\text{unscaled}}(I, T) = \frac{f_I(I)}{\|f_I(I)\|} \cdot \frac{f_T(T)}{\|f_T(T)\|}$$
>
> **(2) Scaled CLIPScore (Typically ~20):** Introduced by Hessel et al. [5]. It applies a ReLU and a $100\times$ scaling factor.
>
> $$\text{CLIPScore}_{\text{scaled}}(I, T) = 100 \cdot \text{ReLU}\left( \frac{f_I(I)}{\|f_I(I)\|} \cdot \frac{f_T(T)}{\|f_T(T)\|} \right)$$
>
> > **[W3] Discrepancy in reported results for prior methods.**
>
> The discrepancies arise from differences in the evaluation pipelines, primarily: (1) prompt preprocessing, and (2) variations in the metric libraries. Details of which were not fully released in prior methods. We re-evaluated all baselines under the following repositories:
> * **Video:** VBench (`Vchitect/VBench`), PSNR/SSIM/LPIPS (`JunyaoHu/common_metrics_on_video_quality`)
> * **Image:** PSNR/SSIM/LPIPS (`chaofengc/IQA-PyTorch`), ImageReward (`zai-org/ImageReward`), CLIPScore (`Taited/clip-score`)
>
> > **[W4] Missing quantitative results.**
>
> Our evaluation follows the standard practices of recent diffusion acceleration literature:
> * FID/FVD and HPS v2: FID is primarily used to evaluate DiT-XL classification models. Recent concurrent works [1][2][3] have similarly not adopted FVD/HPS v2 scores.
> * FLOPs: Please see our response to **Reviewer mtax (Q1)**.
>
> > **[W5] Limited improvements at high speedup ratios (N=7).**
>
> * While the performance gains on HunyuanVideo at extremely high speedup ratios are modest, LearniBridge provides robust SOTA improvements in other models. It achieves a +1.28% VBench score on Wan2.1 at 4.1x, and a +5.7% PSNR on FLUX at 5.87x, representing significant contributions under high acceleration.
> * At extreme ratios (e.g., N=7), severe error accumulation fundamentally limits all related priors. Addressing this bottleneck remains a key direction for future work.
>
> > **[W6] LoRA adapter training still requires significant memory overhead as seen in Appendix A.**
>
> The training memory reported in Appendix A is primarily driven by the base model parameters, which is an unavoidable VRAM cost during standard inference. As shown in Table 5, the LoRA adapters themselves contribute a very lightweight overhead. For example, the base HunyuanVideo DiT takes 25.6 GB, while the attached LoRA adds only 0.588 GB (2.2%).
>
> > **[Q2] What is the layer and timestep used in the analysis of Figure 1? What are the 100 prompts used in this analysis?**
>
> This was conducted on WAN-2.1-1.3B at timestep $t=7$ (total steps $T=50$). The 100 prompts for Fig 2 were sampled from PartiPrompts.
>
> > **[Q3] Pre-calibration prompt selection and potential overlap.**
>
> * Pre-calibration prompts were randomly sampled from *PartiPrompts*. There is no overlap with evaluation benchmarks (*DrawBench* for images and *VBench* for videos). Please see our response to **Reviewer MsPG (Q1)** for details.
> * Prompts used for Table 1: "A group of friends laughing around a campfire at night.", "A child holding sparkler in summer evening.", "Dog chasing leaves swirling in autumn wind.", "A campfire reflecting in calm mountain lake.", "A traveler walking through desert dunes at sunset."
>
> > **[Q4] LoRA training epochs.**
>
> Our calibration set consists of only 3 to 5 samples. Consequently, 1 epoch equals only 3 to 5 training steps. The 500-700 epochs mentioned in Table 5 actually translate to merely 2100-2500 total optimization steps. The entire calibration process takes only 20 minutes, which perfectly aligns with the lightweight and rapid nature of LoRA updates.
>
> **References:**
> [1] Son B, et al. "Relational Feature Caching for Accelerating Diffusion Transformers." *ICLR 2025*.
> [2] Bu J, et al. "DiCache: Let Diffusion Model Determine Its Own Cache." *ICLR 2025*.
> [3] Feng L, et al. "Hicache: Training-free acceleration of diffusion models via hermite polynomial-based feature caching." *ICLR 2025*.
> [4] Radford et al. "Learning Transferable Visual Models From Natural Language Supervision." *ICML 2021*.
> [5] Hessel et al. "CLIPScore: A Reference-free Evaluation Metric for Image Captioning." *EMNLP 2021*.

---

> > ### Author Rebuttal · Reviewer_CVzy · 2026-04-02
> >
> > Thank you for the rebuttal.
> >
> > The concern on novelty still exists as I think that the proposed approach is incremental over the prior methods that already show that predicting intermediate features with cache is feasible in DiTs.
> >
> > The authors still mention the standard CLIPscore formulation and a value of ~0.8 is too high for text-to-image generation tasks. It raises some concerns on the evaluation setup.
> >
> >  Prior methods have reported FID, sFID and IS scores for class-conditional image generation task. A comparison of the proposed method against prior methods on this task would strengthen the paper.
> >
> > The memory overhead should be compared against prior methods.

---

> > > ### Author Response · Authors · 2026-04-08
> > >
> > > We sincerely thank the reviewer for reading our rebuttal and providing these further comments, which we address in detail below.
> > >
> > > > **Q1: The concern on novelty still exists as I think that the proposed approach is incremental over the prior methods that already show that predicting intermediate features with cache is feasible in DiTs.**
> > >
> > > While the feature caching idea was established by DeepCache [1] and forms the common basis of methods like TeaCache [2], TaylorSeer [3], and ToCa [4], our fundamental novelty lies in introducing the first **cache error calibration paradigm**. Specifically, LearniBridge utilizes **LoRA updates** to actively reconstruct future-step representations from cached features, systematically resolving inherent caching errors to achieve up to **5.87x acceleration**.
> > >
> > > > **Q2: The authors still mention the standard CLIPscore formulation and a value of ~0.8 is too high for text-to-image generation tasks. It raises some concerns on the evaluation setup.**
> > >
> > > As detailed in Section 4.1, we evaluated FLUX.1-dev on the DrawBench benchmark using the standard `Taited/clip-score` toolkit.
> > > A score of ~0.8 is completely within the normal range for this model, as corroborated by existing literature [5], which reports scores around ~0.9.
> > >
> > > > **Q3: Prior methods have reported FID, sFID and IS scores for class-conditional image generation task. A comparison of the proposed method against prior methods on this task would strengthen the paper.**
> > >
> > > Quantitative comparison on class-to-image generation on ImageNet with DiT-XL/2.
> > >
> > > | Method | FLOPs (T) $\downarrow$ | Speed $\uparrow$ | FID $\downarrow$ | sFID $\downarrow$ | Inception Score $\uparrow$ |
> > > | :--- | :--- | :--- | :--- | :--- | :--- |
> > > | DDIM-50 steps | 23.74 | $1.00\times$ | 2.32 | 4.32 | 241.25 |
> > > | DDIM-10 steps | 4.75 | $5.00\times$ | 12.15 | 11.33 | 159.13 |
> > > | DDIM-8 steps | 3.80 | $6.25\times$ | 23.13 | 19.23 | 120.58 |
> > > | ToCa ($N=9$) | 6.34 | $3.75\times$ | 6.55 | 7.10 | 189.53 |
> > > | TaylorSeer ($N=6, O=4$) | 4.76 | $4.98\times$ | 3.11 | 6.35 | 223.85 |
> > > | **LearniBridge ($N=6$)** | **4.81** | **$4.93\times$** | **2.98** | **6.29** | **225.78** |
> > > | ToCa ($N=13$) | 4.03 | $5.90\times$ | 21.24 | 19.93 | 116.08 |
> > > | TaylorSeer ($N=8, O=4$) | 3.82 | $6.22\times$ | 4.40 | 7.34 | 205.00 |
> > > | **LearniBridge ($N=8$)** | **3.84** | **$6.17\times$** | **4.31** | **6.99** | **206.12** |
> > >
> > > On the task of class-conditional ImageNet generation with the DiT-XL/2 model, LearniBridge again demonstrates consistent improvement.
> > >
> > > > **Q4: The memory overhead should be compared against prior methods.**
> > >
> > > We compare the peak GPU memory consumption of LearniBridge against prior methods using the HunyuanVideo model at a resolution of $544 \times 960 \times 49$ (height $\times$ width $\times$ frames), as summarized in the table below.
> > >
> > > | Method | Memory Usage (MiB) |
> > > | :--- | :--- |
> > > | Original | 32,990 |
> > > | TeaCache | 34,238 |
> > > | **LearniBridge (Ours)** | **34,798** |
> > > | Taylorseer | 67,582 |
> > >
> > > As shown in the table, LearniBridge introduces only a negligible memory overhead compared to the original model.
> > >
> > > **Reference**
> > > [1] Ma, Xinyin, Gongfan Fang, and Xinchao Wang. "Deepcache: Accelerating diffusion models for free." *CVPR 2024.*
> > > [2] Zhao, Zhi, et al. "Timestep Embedding Tells: It's Time to Cache for Video Diffusion Model." *CVPR 2025.*
> > > [3] Liu, Qilin, et al. "From Reusing to Forecasting: Accelerating Diffusion Models with TaylorSeers." *ICCV 2025.*
> > > [4] Yang, Xingkang, et al. "Accelerating Diffusion Transformers with Token-wise Feature Caching." *ICLR 2025.*
> > > [5] Ertacache: Error rectification and timesteps adjustment for efficient diffusion. X Peng, C Yan, H Liu, R Ma, F Chen, X Wang, Z Wu, S Liu, M Lin. *ICLR 2026.*

---

### Official Review · Reviewer_MsPG · 2026-03-13

**Soundness:** 3
**Presentation:** 2
**Significance:** 3
**Originality:** 3
**Overall Recommendation:** 4
**Confidence:** 4

**Summary:**

This paper aims to improve the generation quality of diffusion models in acceleration regimes based on feature caching. Specifically, the authors fine-tune the final DiT block with 10 individual LoRAs, motivated by an SVD-based rank analysis claiming that the cross-timestep residuals are prompt-invariant and temporally consistent. The training phase only requires 3-5 prompts and their full diffusion trajectories, which is relatively efficient. Experiments on multiple text-to-image and text-to-video benchmarks demonstrate that the proposed method improves generation quality across different acceleration ratios.

**Compliance With Llm Reviewing Policy:**

Affirmed.

**Final Justification:**

The rebuttal removes most of my concerns and I maintain my score.

**Key Questions For Authors:**

Please address Points 1 and 2 in the "Weaknesses".

**Limitations:**

Yes

**Strengths And Weaknesses:**

Strengths:
1. Improving image and video diffusion models at high acceleration ratios is valuable.
2. The low-rank motivation presented in Section 3.2 is clever and insightful.
3. The experimental results are consistently competitive across benchmarks.

Weaknesses or questions:
1. Regarding the low-rank structure, which is claimed to be **prompt-invariant**:
- I did not find any experiments or ablation studies detailing how the set of 3-5 prompts is selected.
- Are they chosen randomly or empirically curated? Are simple prompts as helpful as complex ones? Would increasing the number of prompts further improve or hurt the performance?
2. Regarding the low-rank structure, which is claimed to be **temporally consistent**:
- Eq. 14: How many $(t, t-k)$ pairs are there for a certain $T$ and $N$? I assume that the same 10 LoRAs are applied to every $(t, t-k)$ pair. It is somewhat surprising that a single set of LoRA parameters can handle all cross-timestep adjustments, particularly given that the input $x_t^L$ is directly modulated by AdaLN using the previous $t$. Experimental validation or discussion of this would be appreciated.
- Appx B.1: The figures show $t \in [7, 47]$. However, which specific diffusion model is used here? What is the exact total timestep $T$? Does this property only hold at small timesteps?
3. The writing can be improved:
- Section 2.2: "ToCa" is introduced without a citation at its first appearance.
- Section 2.2: Incorrect left quotation marks are used before "replication" and "prediction".
- Section 3.1, Feature Caching: The procedure described here is somewhat confusing and is not consistent with the proposed method.
- Section 3.2-3.3: The notations (e.g., $f^l$, $\Delta W^l$) and LoRA approximations should be layer-wise (i.e., the 10 layers in Figure 1: SA's QKVO, CA's QKVO, and FFN's 1&2). However, DiT's block-level notations like $F$ are frequently mixed in, causing inconsistency and confusion.
- Figure 1: The label "ffn.0" should be "ffn.1" for consistency.

---

> ### Author Rebuttal · Authors · 2026-03-31
>
> We thank you for recognizing the effectiveness of our acceleration framework and our low-rank approach. We embrace your constructive feedback as a valuable means to enhance the paper’s quality. To that end, we provide the following clarifications and responses to your inquiries:
>
> > **[Q1] Regarding the prompt-invariant low-rank structure, how are the 3–5 prompts selected? Are they randomly sampled or curated, and how do prompt complexity and prompt number affect performance?**
>
> Our calibration prompts are randomly sampled from *PartiPrompts* (zero overlap with *DrawBench* for images and *VBench* for videos) and extended legth via gpt-5.2. We empirically found that using these semantically rich, medium-to-long prompts yields significantly more robust calibration than using simpler inputs.
> To systematically validate this design choice, we conducted comprehensive ablation studies on FLUX T2I task (evaluating 200 DrawBench images, with $N=5$ and $r=64$, as Sec 4.2 in paper).
>
> **(1) Effect of Prompt length:**
> We categorized prompt length by word count: short (0–5), medium (10–15), and long (20–25).
>
> | Prompt length | PSNR ↑ | SSIM ↑ | LPIPS ↓ |
> | :--- | :--- | :--- | :--- |
> | Short (0–5 words) | 29.6247 | 0.6927 | 0.3135 |
> | **Medium (10–15 words)** | **30.1569** | **0.7898** | **0.2654** |
> | **Long (20–25 words)** | **30.0137** | **0.7878** | **0.2594** |
>
> The method performs better with medium and long prompts, while short prompts lead to degraded performance, suggesting that sufficient semantic richness is important. Based on this observation, we constrain GPT-5.2 prompt rewriting to produce medium-to-long prompts.
>
> **(2) Effect of Prompt Number:**
> We eval number on the prompt selection scheme.
>
> | Number of Prompts | PSNR ↑ | SSIM ↑ | LPIPS ↓ |
> | :--- | :--- | :--- | :--- |
> | 1 | 27.7913 | 0.6771 | 0.3829 |
> | 3 | 29.1079 | 0.6970 | 0.3184 |
> | **5 (Default)** | **30.1623** | **0.7799** | **0.2681** |
> | 10 | 30.0728 | 0.7765 | 0.2635 |
> | 20 | 30.2297 | 0.7679 | 0.2746 |
> | 30 | 30.0814 | 0.7642 | 0.2699 |
>
> Performance improves significantly from 1 to 5 prompts, effectively covering the necessary semantic variance. Beyond 5 prompts, the gains saturate with only minor fluctuations, indicating diminishing returns. Therefore, 5 prompts perfectly strike the optimal balance between calibration efficiency and generative quality.
>
> > **[Q2] Construction of $(t, t-k)$ Pairs, LoRA Assignment, and Generalization.**
>
> **(1) Construction of $(t, t-k)$ pairs.**
> Given a diffusion trajectory of length $T$ and a timestep interval $N$, $(t, t-k)$ denotes a pair of timesteps within the same segment, where $t$ is a earlier timestep and $t-k$ is an later timestep with $k = N - 1$. This pair is used to train a LoRA adapter to map the cached input $x_t^L$ to the corresponding output $F^L(x_{t-k})$.
>
> For example, when $T = 50$ and $N = 5$, the resulting $(t, t-k)$ pairs are:
>
> $$
> (47,43), (42,38), (37,33), (32,28), (27,23), (22,18), (17,13), (12,8), (7,3), (2,0)
> $$
>
> **(2) LoRA parameterization across timestep pairs.**
> To clarify, by the phrase "temporally consistent" in our manuscript, we simply meant that the **low-rank structure holds valid across different timesteps**. Accordingly, the LoRA parameters are **not shared globally across all timestep pairs**. Instead, we adopt a segment-wise parameterization strategy:
> - Each timestep segment is associated with one independent LoRA module
> - In the above example, this results in 10 distinct LoRA modules
> - During inference, the corresponding LoRA is selected and applied based on the timestep interval
>
> **(3) Model setting and generalization (Regarding Appx B.1).**
> All experiments in this section are conducted on WAN-2.1-1.3B with $T = 50$. The method remains effective for larger timestep intervals (e.g., $k = 7$), indicating robustness across temporal scales. We will explicitly clarify these details and include the additional results in the revised Appx B.1.

---

> > ### Author Rebuttal · Reviewer_MsPG · 2026-04-02
> >
> > I thank the authors for the detailed rebuttal, which has addressed most of my concerns. However, I think the "segment-wise parameterization strategy" is not elegant enough, as it requires one independent set of LoRA modules (10 linear layers) for each segment, which restricts the flexibility for different $T$ and acceleration ratio $N$ settings.
> > Therefore, I lean towards acceptance, but I still believe the paper has some weaknesses and limitations.

---

> > > ### Author Response · Authors · 2026-04-08
> > >
> > > We sincerely thank the reviewer for the engaging discussion, the valuable feedback throughout the rebuttal process, and the support for our work. The insightful comment points out a highly promising direction for our future work: exploring continuous or dynamic parameterization strategies to improve flexibility across different acceleration settings.

---

### Decision · Program_Chairs · 2026-04-30

**Decision:**

Accept (regular)

**Comment:**

The paper presents a well-motivated and practically relevant approach to accelerating diffusion transformers via learnable calibration of feature caching, leveraging an intriguing inherent low-rank structure via lightweight LoRA-based mechanism. Reviewers consistently highlight its technical soundness, clarity, and strong empirical performance across multiple image and video benchmarks, demonstrating meaningful quality preservation under significant speedups. While some concerns remain regarding novelty and aspects of evaluation, these are relatively minor compared to the overall contribution. The work offers a clean and effective perspective on improving acceleration and is likely to inspire further research in this direction.